# Multi-scale modeling of macrophage—T cell interactions within the tumor microenvironment

**Colin G. Cess**[1], **Stacey D. Finley**[1,2,3]*

**1** Department of Biomedical Engineering, University of Southern California, Los Angeles, California, United States of America, **2** Department of Quantitative and Biological Sciences, University of Southern California, Los Angeles, California, United States of America, **3** Mork Family Department of Chemical Engineering and Materials Science, University of Southern California, Los Angeles, California, United States of America

* sfinley@usc.edu

**Data Availability Statement:** Github: https://github.com/FinleyLabUSC/Early-TME-ABM-PLOS-Comp-Bio.

**Funding:** This work was supported by the USC Viterbi/Graduate School Merit Fellowship to CGC

## Abstract

Within the tumor microenvironment, macrophages exist in an immunosuppressive state, preventing T cells from eliminating the tumor. Due to this, research is focusing on immunotherapies that specifically target macrophages in order to reduce their immunosuppressive capabilities and promote T cell function. In this study, we develop an agent-based model consisting of the interactions between macrophages, T cells, and tumor cells to determine how the immune response changes due to three macrophage-based immunotherapeutic strategies: macrophage depletion, recruitment inhibition, and macrophage reeducation. We find that reeducation, which converts the macrophages into an immune-promoting phenotype, is the most effective strategy and that the macrophage recruitment rate and tumor proliferation rate (tumor-specific properties) have large impacts on therapy efficacy. We also employ a novel method of using a neural network to reduce the computational complexity of an intracellular signaling mechanistic model.

## Author summary

We present a multi-scale agent-based model of macrophages and T cells within the tumor microenvironment. To increase the biological detail, we include an intracellular mechanistic model in the macrophages, employing a method of using neural networks to reduce the mechanistic model into a simple input/output model. With the mechanistic model, we are able to predict the effects of specifically inhibiting a part of the intracellular signaling pathway, as opposed to just making phenotypic predictions. Using the integrated modeling framework, we are able to predict the impacts of immunosuppressive macrophages on T cell function and predict how macrophage-based immunotherapies can reduce immunosuppression. Altogether, we present a useful framework for studying cell-cell interactions in the tumor microenvironment and the effects of immune cell-targeting therapies.

(http://graduateschool.usc.edu) and the USC Center for Computational Modeling of Cancer (http://modelingcancer.usc.edu). The funders had no role in study design, data collection and analysis, decision to publish, or preparation of the manuscript."

**Competing interests:** The authors have declared that no competing interests exist.

## Introduction

A key feature of the tumor microenvironment (TME) is that the normal immune response, which should be able to target and kill malignant cells, is dysfunctional [1,2]. Specifically, some immune cells are suppressed and unable to carry out their functions, while other immune cells are corrupted into a pro-tumor state and actively work to increase tumor growth. The dysfunctional immune response is a common feature across tumor types, and it is thought that tumors develop only after evading the immune system [3]. Recently, immunotherapies have been developed in attempts to reactivate the immune system so that it can carry out its normal function and remove the tumor [4–6]. Although the efficacy of immunotherapy has increased in recent years, the use of immunotherapeutic strategies to treat solid tumors has been largely unsuccessful [7–9]. Therefore, further examination of the immune suppressing mechanisms within the TME is needed in order to develop more effective immunotherapies.

One of the most common, and most influential, types of immune cell in the TME is the tumor-associated macrophage (TAM) [10,11]. Macrophages have various roles within the normal immune response, having cytotoxic capability and the ability to present antigens to T cells [10]. Depending on environmental signals, macrophages can display a variety of phenotypes, ranging from pro-inflammatory and immune-supporting to immunosuppressive with wound-healing properties. While macrophage phenotype exists on a spectrum [12], for conceptual purposes it is divided into two main states: M1 (immune-promoting) and M2 (immunosuppressive). Due to influence from the tumor, most TAMs are in an M2-like state and further promote tumor growth. TAMs are able to promote tumor cell proliferation, induce angiogenesis, enable tumor cell migration and metastasis, and suppress the function of anti-tumor immune cells [10,11,13,14]. Due to their important roles in tumor growth, TAMs have become the subject of various immunotherapies [9,15,16]. These treatment strategies aim to either reduce the number of TAMs within the TME, which would limit their suppression of T cells, or convert TAMs into an M1-like state, which would enhance T cell function.

T cells are considered to be the main portion of the adaptive immune system for eliminating tumors, being able to detect tumor-associated antigens and then kill tumor cells [17–19]. It is hypothesized that most tumors are eliminated by T cells early on and that only a small number of tumors manage to escape and go on to have clinical significance [3]. As tumors grow further, T cell function is suppressed, diminishing the immune response [17–20]. Cytotoxic T cells (CTLs) can become exhausted, having limited proliferative and cytotoxic function, due to excessive stimulation and the expression of checkpoint proteins such as CTLA-4 and PD-L1 on M2 macrophages and tumor cells [21]. The tumor also increases the excretion of chemokines that attract regulatory T cells (Tregs) and cytokines from that promote the conversion of T helper (Th) cells into Tregs, which assist in suppressing CTL function [22]. M2 macrophages can also produce these Treg-promoting cytokines.

Various immunotherapies have been developed to reactivate the immune system and promote tumor removal. Checkpoint inhibition aims to block ligands such as CTLA-4 and PD-L1 on tumors and M2 macrophages [4–7,20]. Blocking these ligands has been shown to increase T cell expansion at the tumor site and promote the killing of tumor cells, though it is unclear whether this is due to the restoration of the functions of T cells already at the tumor site or due to the infiltration of new T cells, as evidence has been found for both [20,23]. Another method of immunotherapy is adoptive T cell therapy, in which T cells are removed from the patient, expanded *ex vivo*, and then given back to the patient in an attempt to boost the immune response [5,6,24]. Sometimes the T cells are modified in order to better detect tumor cells, as is the case with chimeric antigen receptor (CAR)-engineered T cells. While immunotherapy is successful in some cases, particularly in hematological malignancies, it often fails in solid

tumors. This lack of success indicates that there is a complex interplay within the cells residing in the TME that prevents immune function.

Many computational models have been developed to better understand interactions between tumor and immune cells. We recently reviewed mathematical models of tumor-immune interactions across various scales [25]. These models aim to both examine general tumor-immune behavior and to test the effects of immunotherapy. Significant modeling efforts have focused on T cell-mediated killing. For example, Gong et al. developed a model of how PD-L1 expression impacts T cell response [26]. They also examined how tumor mutational burden and antigen strength, which impact the strength of the T cell response, influence tumor growth. Kather et al. examined T cell response in relation to tumor stroma, which physically inhibits both immune cell infiltration and tumor growth [27]. They determined that a high level of stroma slows tumor growth when the number of T cells is low but prevents immune cell-mediated elimination of the tumor when there is a high number of T cells. Therefore, combining therapies that increase T cell count with therapies that reduce tumor stroma could increase T cell infiltration and tumor removal. Other studies have focused on lymph node dynamics and the effects of tumor proliferation, antigenicity, and T cell recruitment on tumor removal [28].

Mathematical modeling has also been applied to study the role of macrophages in tumor elimination. Some of these models have focused on the interplay between M1 and M2 macrophages at the early stages of tumor growth. Both Wells et al.[29] and Malbacher et al.[30] developed models where macrophages can differentiate into either an M1 or an M2 phenotype based on factors secreted by the growing tumor. These macrophages then either inhibit or promote tumor growth, depending on their phenotype. El-kenawi et al. focused on the effects of tumor-induced acidity on macrophage differentiation, allowing macrophages to differentiate on a spectrum as opposed to discrete phenotypes, to more accurately represent macrophage state [31]. Other models have focused on interactions between macrophages and tumors cells that lead to increased tumor migration and metastatic potential due to paracrine signaling loops between the two cell types [32–34].

While there are examples of mathematical models that consider multiple types of immune cells in the local tissue microenvironment [35], most models do not consider multiple types of immune cells in the TME. However, it is important to account for the various immune cell populations since immune cells interact with each other directly and via diffusible signaling factors. Therefore, in this study, we focus on the interactions between macrophages and T cells. Using an agent-based model (ABM), we examine the growth of a micrometastasis and how macrophage differentiation affects the ability of the T cells to eliminate the tumor. We also model the effects of three macrophage-based immunotherapies (macrophage depletion, recruitment inhibition, and macrophage reeducation) and investigate how differing rates of tumor proliferation and macrophage recruitment affect the efficacy of each treatment. To model macrophage differentiation, we employ a mechanistic intracellular signaling model within the macrophages. In order to improve computational time, we use a neural network to predict the mechanistic model outputs based on signaling inputs from the TME. We find that macrophage reeducation is the most powerful of the three immunotherapies simulated due to the promotion of T cell function, and that tumor proliferation rate and macrophage recruitment rate can have large impacts on the efficacy of therapy.

## Results

### Model construction

As explained in detail in the methods section, we have constructed a multi-scale agent-based model of the interactions between macrophages, T cells, and tumor cells, along with the

cytokines IL-4 and IFN-γ. The model represents a 2D tissue slice, and cells are constrained to a lattice, with one cell per lattice site. Briefly, macrophages are initially present within the tissue and more are recruited to the tumor site as simulation progresses. They are able to differentiate based on cytokine concentration into either an M1-like or M2-like state. Differentiation is based on an intracellular signaling model, which was then reduced to a simple input/output model using a neural network to relate cytokine concentrations to differentiation. T cells are recruited based on tumor cell death, and act to kill tumor cells. They become fully active upon antigen contact, allowing them to kill tumor cells and secrete IFN-γ. Activation is promoted by M1 macrophages and inhibited by M2 macrophages. Tumor cells proliferate and secrete IL-4. With this model, we predict the effects of three macrophage-based immunotherapies (macrophage depletion, recruitment inhibition, and macrophage reeducation via PI3K inhibition) and how they impact the ability of T cells to remove the tumor. A model schematic along with an example spatial distribution of cells is shown in Fig 1. From the spatial distribution (Fig 1B), which shows a sample simulation without treatment, we see that macrophages surround the tumor, primarily in the M2 state, preventing some T cells from reaching the tumor and inhibiting T cells that are recruited to the tumor site. There are active T cells immediately adjacent to tumor cells; however, there are not enough to eliminate the tumor.

## Model behavior without treatment

Prior to simulating macrophage-based interventions, we ran simulations without any treatment to have a reference for comparing the efficacy of different treatment strategies. To understand how various parameters relating to the immune response affect model behavior, we sampled over a wide range of values above and below our base parameter values using Latin Hypercube Sampling (LHS), generating a total of 500 parameter sets. Due to the stochastic behavior of the model, each parameter set was simulated 100 times in order to obtain average behavior. We note that LHS generates a series of parameter sets where each parameter value only appears once. Therefore, the sharp spikes seen in the results are due to each parameter

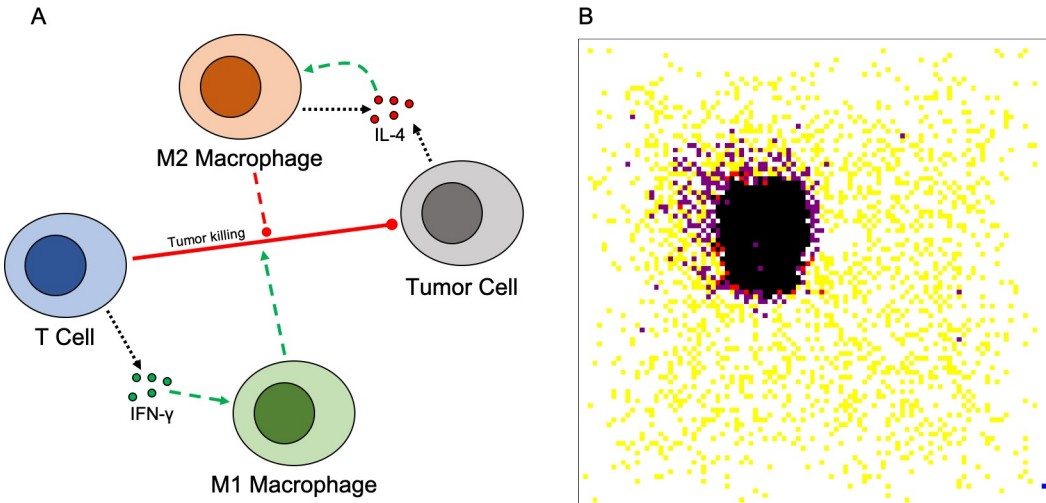

**Fig 1. Model schematic and representative simulation result.** (A) Model schematic. T cells secrete IFN-γ, which promotes M1 differentiation. Tumor cells and M2 macrophages secrete IL-4, which promotes M2 differentiation. T cells kill tumor cells. M1 macrophages promote T cell function, while M2 macrophages inhibit it. (B) Representative spatial distribution of cells once the tumor reaches the equilibrium state. Tumor cells (black), T cells (yellow), active T cells (red), M0 macrophages (blue), M2 macrophages (purple). There are no M1 macrophages present at the end of this simulation.

value being simulated once, and not the average response at the parameter value. Despite this, we still see trends for certain parameters. We found that macrophage recruitment rate had the only noticeable effect on the fraction of tumors removed by the immune system. Specifically, higher recruitment rates, and thus more macrophages in the system, lead to a lower fraction of tumors that were removed (S1 Fig). We also found that macrophage recruitment rate and lifespan were correlated to the final number of tumor cells at the end of simulation (S2 Fig). That is, having more macrophages, which would be in the M2 state (discussed in detail below), present in the environment, due to either increased recruitment or longer lifespan, leads to more tumor cells at the end of simulation. These results support the finding that high numbers of TAMs have a worse clinical outcome [15]. This analysis also supports our focus on predicting the effects of macrophage-based therapies. The final values of the model parameters were set so that few tumors were removed by the immune system without treatment, so that tumor killing in subsequent simulations would be due primarily to treatment, and to match those used in similar models.

We used the base model to explore the dynamics of each cell type and diffusible factor present in the TME. Shown in Fig 2 are the time courses for 100 simulations over 200 days without treatment. For all simulations where the immune system failed to eliminate the tumor (99%), each cell type reached an equilibrium state. This state can be considered the "immune control" phase of tumor growth, which is thought to take place over several years and involves the selection of tumor cells that are resistant to the immune system [3]. We conclude that this equilibrium state is indeed due to the cytotoxic function of the T cells and not spatial inhibition, with S3 Fig comparing tumor growth curves in the absence of immune cells and with immune cells present but without function, finding these two curves to be almost identical. After an initial drop due to the introduction of T cells to the environment, the tumor cell population evens out at around 350 cells (Fig 2A). The naive macrophage population drops immediately and stays at a low number due to continuous recruitment to the TME and subsequent differentiation (Fig 2B). In this set of simulations, we see no macrophage differentiation to the M1 phenotype (Fig 2C) whereas the number of M2 macrophages reaches a high level (Fig 2D). The time courses for the total number of T cells (Fig 2E) resembles a delayed version of those for tumor cells. The time courses for active T cells follows this, though at very low levels (Fig 2F). Due to the number of tumor cells, average and maximum IL-4 levels are relatively high (Fig 2G and 2H) while IFN-γ levels (Fig 2I and 2J) are low due to low T cell activation. This difference causes the absence of M0 differentiation into the M1 phenotype. We also briefly examined how initial macrophage density impacts tumor dynamics (S4 Fig). We found that while this impacts initial tumor dynamics, long-term dynamics were not impacted by initial macrophage count.

## Effects of continuous treatment

After evaluating tumor growth in the absence of treatment, we evaluated the efficacy of three macrophage-based therapies: (1) macrophage depletion, (2) inhibition of macrophage recruitment, and (3) re-education of macrophages via PI3K inhibition. To initially explore how each treatment strategy affects the immune response, we simulated each treatment at ten different strengths. Treatment was started 100 days after tumor initiation, which was after the system had reached equilibrium. The treatment continued until either the tumor was removed or until 200 days of simulation time was reached. We repeated each simulation 100 times and averaged the results to obtain the general effect of each treatment. A primary output of these simulations is the fraction of tumors removed—the number of tumors eliminated after treatment was started. We also calculated the time from start of therapy to tumor elimination and

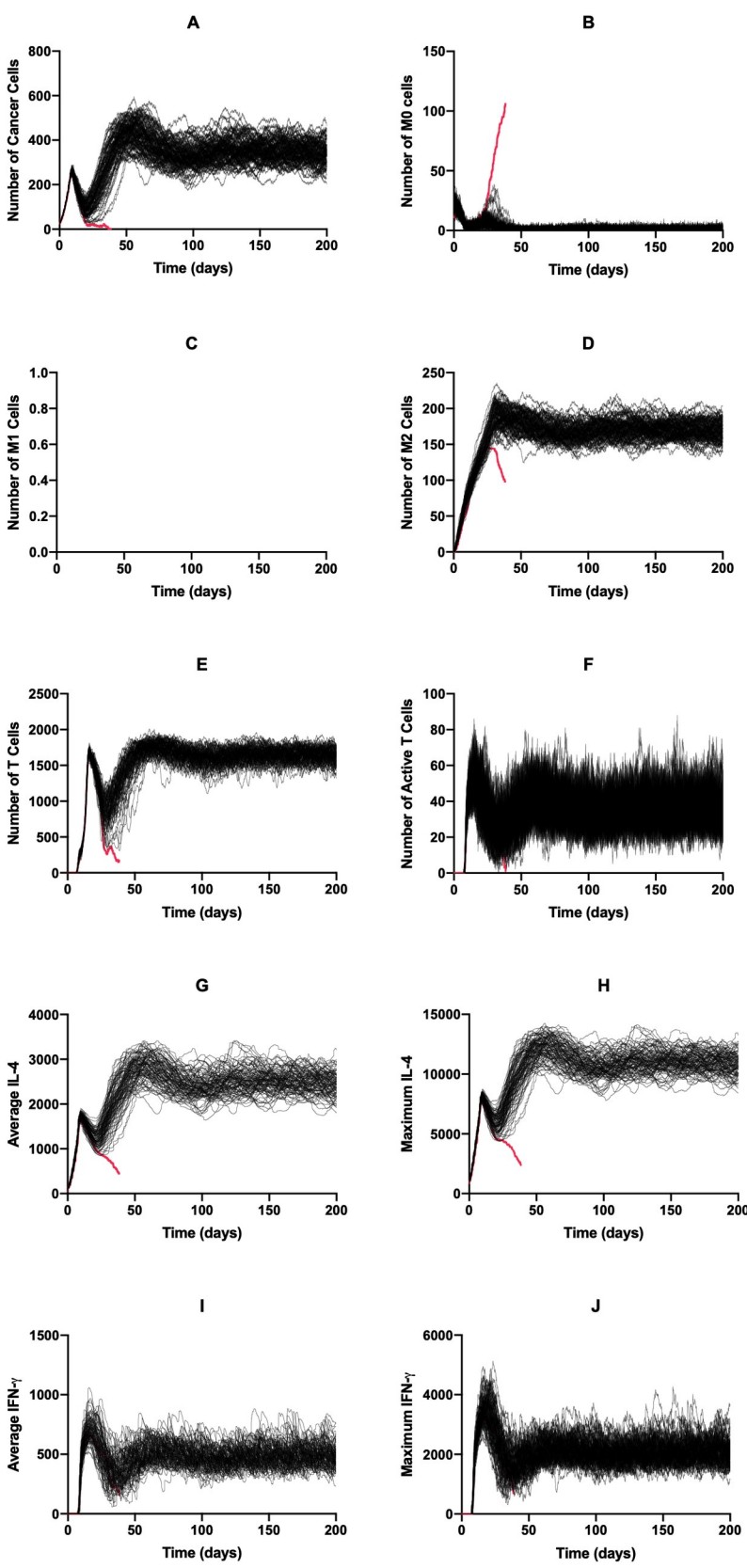

**Fig 2. Time courses with baseline parameters and no treatment.** Time courses for tumors that were not removed by the immune system are shown in black; those for tumors that were removed are shown in red. There is only one simulation here that led to tumor removal. (A) Cancer cells, (B) M0 cells, (C) M1 cells, (D) M2 cells, (E) total T cells, (F) active T cells, (G) average IL-4, (H) maximum IL-4, (I) average IFN-γ, (J) Maximum IFN-γ. (G)–(J): units are number of molecules.

the maximum numbers of M1 macrophages, T cells, and active T cells. Because treatment starts once equilibrium is reached, the system is already at the maximum number of tumor cells and M2 macrophages, so we do not record their maximum values during treatment.

First, we investigated the effects of macrophage depletion (Fig 3A). The "Depletion Probability" displayed on the *x*-axis is the probability that a macrophage has of being removed at each time step, with an equal probability for each macrophage phenotype. Fig 3A-i shows the percent of tumors eliminated after treatment is started. We see that there is a drastic jump from almost no effectiveness at a probability of 0.001 to almost complete tumor removal at a probability of 0.002, which then increases slightly and stays at complete tumor removal as the probability is increased. It is interesting that there is such a sharp increase in effectiveness without a corresponding increase in the maximum number of M1 macrophages or active T cells. We believe that this increase is due to removal of enough macrophages around the tumor to allow T cells near the tumor to infiltrate better and remove the tumor. Following this, the average number of days needed to remove the tumor decreases and then remains static (Fig 3A-ii). We see that the maximum number of M1 macrophages increases with depletion probability to a point, then begins to decrease at the highest depletion probabilities (Fig 3A-iii). We believe that the increase is due to the decrease in IL-4 as tumor cells and M2 macrophages are removed, allowing newly recruited macrophages to differentiate into the M1 state. In S5 Fig we show the time courses at a depletion probability of 0.006, which shows a decrease in IL-4 and and increase in IFN just prior to the increase in M1 macrophages. As depletion probability further increases, these new macrophages are removed fast enough to decrease the maximum number of M1 macrophages. However, due to the low numbers of M1 macrophages and large error bars, we cannot make any definitive inferences from this. The total number of T cells increases very slightly with depletion probability (Fig 3A-iv) while T cell activation sees a more drastic increase (Fig 3A-v) corresponding with the number of M1 macrophages.

Examining the time course for a depletion probability of 0.002 (S6 Fig), we find that in simulations that fail to remove the tumor, the tumor cell population reaches a new equilibrium state that is lower than the original (S6A Fig). As the tumor is removed, there is an increase in the number of naïve macrophages (S6B Fig) due to less of an influence from the tumor for macrophage differentiation. At this level of depletion, no M1 macrophages arise during simulation (S6C Fig), while the number of M2 macrophages decreases (S6D Fig). We also see little difference in T cell and active T cell dynamics (S6E and S6F Fig) between simulations that removed the tumor and those that did not, presumably due to the lack of M1 macrophages, which promote T cell activation.

The second treatment strategy we employed was inhibition of macrophage recruitment, which is similar to macrophage depletion in that it eliminates macrophages regardless of phenotype. On the *x*-axes for Fig 3B is the "Inhibition Strength," which is the fraction that the parameter for recruitment rate was reduced by. We see that below 0.6 inhibition, treatment is ineffective in removing the tumor. At 0.6 there is very minor effectiveness, which then rapidly increases with inhibition strength (Fig 3B-i). As inhibition strength increases, we see a slight decrease in the average time needed to remove the tumor (Fig 3B-ii), however it does take longer to remove the tumor than with macrophage depletion probabilities that achieved similar tumor removal (compare Fig 3A-ii and Fig 3B-ii). Whereas macrophage depletion showed

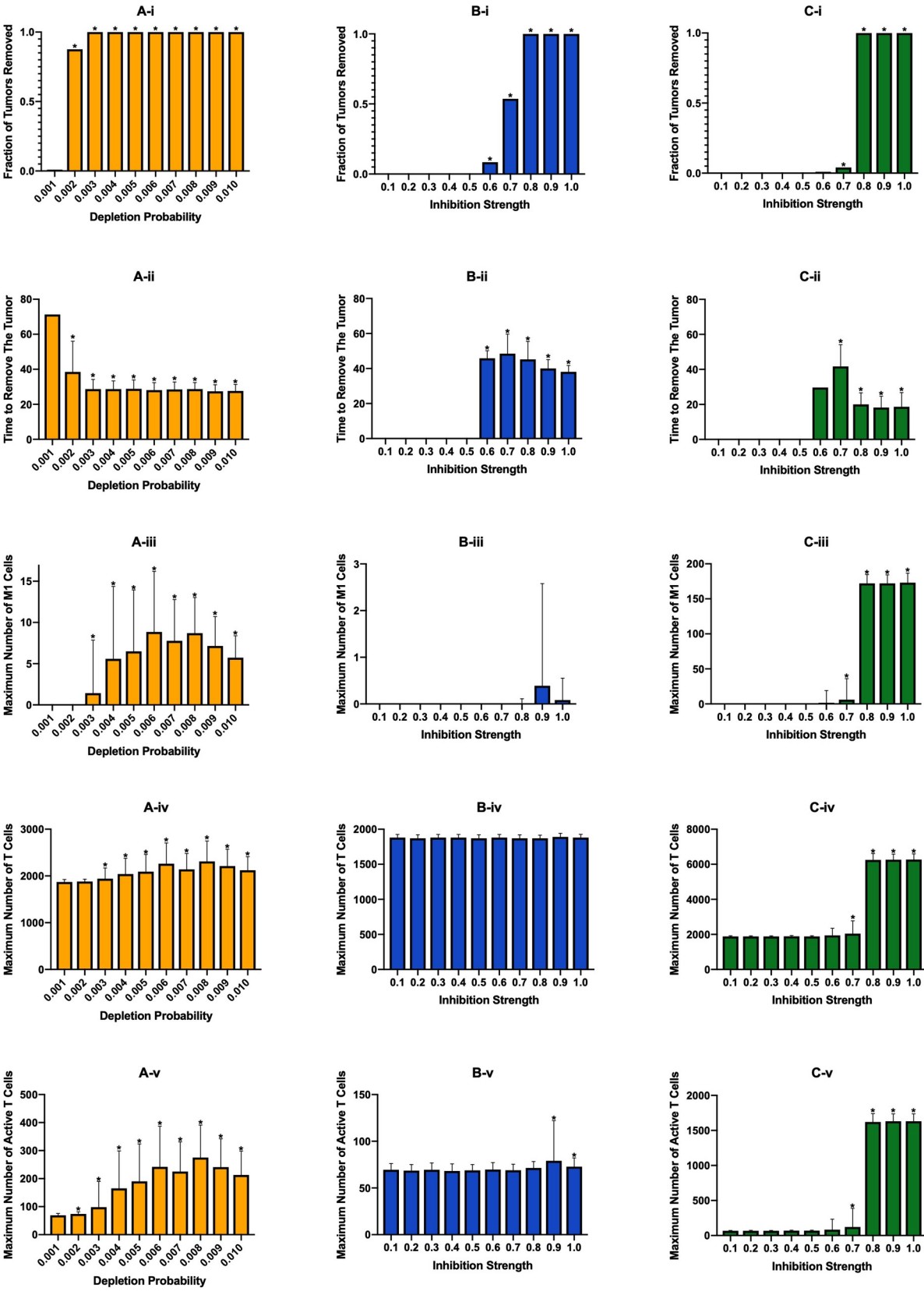

**Fig 3. Effects of continuous immunotherapy started at 100 days of simulation.** (A) Macrophage depletion, (B) recruitment inhibition, and (C) PI3K inhibition. (i) fraction of tumors removed after starting therapy. (ii) time (days) from starting treatment to tumor removal. It is averaged over the 100 simulations and is equal to zero if no tumors were removed at that treatment level. (iii) maximum number of M1 macrophages. (iv) maximum number of total T cells. (v) maximum number of active T cells. Note the differences in y-axis scales across treatment strategies. Asterisks signify that a result is statistically significant (p<0.01) from the result of the lowest treatment strength. We note that for the time needed to remove the tumor (ii), we plot the time averaged over only simulations where the tumor was removed. Therefore, while some bars may appear much higher than that of the lowest treatment strength, they only represent a small number of simulations out of 100 and thus were not found to be statistically significant.

some increase in the number of M1 macrophages, they are essentially nonexistent with recruitment inhibition (Fig 3B-iii). We believe this is due to the fact that macrophage depletion removes macrophages from the environment while recruitment inhibition prevents new macrophages from entering. While both methods lower the total number of macrophages, the macrohpages that remain even with recruitment inhibition are closer to the tumor, where IL-4 levels are highest, which keeps them in the M2 state. With macrophage depletion, new macrophages enter the environment farther from the tumor where IL-4 levels are lower, allowing some of them to differentiate to the M1 state. With this, we see that there is no change in the numbers of total or active T cells (Fig 3B-iv,v), with T cell activation being fairly low.

Examining the time courses for a recruitment inhibition of 0.7, which eliminated roughly half of the tumors, we find that the time courses for tumors that were removed are very similar to those that were not (S7 Fig). The dynamics for the tumor cells are very similar for the initial decrease, with tumors that were eliminated continuing to decrease, while tumors that were not suddenly leveling off to a new equilibrium (S7A Fig). We believe this is due to inherent randomness and M2 macrophages preventing T cells from advancing t the tumor. We do see some increase in naive macrophages in tumors that were eliminated (S7B Fig). Like the simulations without treatment, no M1 macrophages appear (S7C Fig). The numbers of M2 macrophages, total T cells, and active T cells exhibit similar behavior as the tumor cells (S7D–S7F Fig). IL-4 and IFN-γ concentrations mirror tumor cell and active T cell dynamics, respectively (S7G–S7J Fig).

The final treatment strategy, PI3K inhibition, yielded similar efficacies as recruitment inhibition, though with a steeper shift from non-effectiveness to effectiveness (Fig 3Ci). However, PI3K inhibition displayed the fastest tumor removal time (Fig 3Cii) due to increases in M1 macropahges and thus T cell function. As intended, because PI3K inhibition aims to reeducate the macrophages, the number of M1 macrophages is much higher with this strategy (Fig 3Ciii). This increase correlates with an increase in the number of total and active T cells (Fig 3Civ,v). We also examined the time courses for a PI3K inhibition of 0.8 (S8 Fig). There is the expected decrease in tumor cells (S8A Fig) and, since this treatment does not remove macrophages, there is an increase in naive macrophages (S8B Fig) due to a decreased pressure to differentiate as the number of cancer cells is reduced. The number of M1 macrophages rapidly increases immediately after treatment, while the number of M2 macrophages rapidly decreases (S8C and S8D Fig). Both total and active T cells increase greatly with the increase in M1 macrophages (S8E and S8F Fig) due to the influence of the M1 macrophages. As with other treatments, IL-4 and IFN-γ levels follow tumor and T cell dynamics (S8G–S8J Fig).

Overall, we find that, at the baseline parameters, treatment efficacy for all three strategies steeply increases from ineffective to very effective when simulated continuously. PI3K inhibition, which converts the macrophages to the M1 state, leads to higher levels of T cell activation and removes the tumor at a faster rate than the other two treatments. Interestingly, although both of these treatments reduce the number of macrophages in the TME, we see that macrophage depletion leads to a slight increase in M1 macrophages while recruitment inhibition does not.

## Treatment cycles

While it is ideal to be able to give a continuous treatment, this is unrealistic due to various reasons, such as patient compliance, inability to give treatment continuously due to pharmaceutical implementation, or the desire to give as little treatment as possible in order to avoid potential side-effects. Therefore, we ran several sets of simulations where treatment was cycled on and off. We vary both the total length of the treatment cycle and the amount of time during the cycle that treatment is given. We performed these simulations only for higher treatment strengths, as it can be reasoned that cycling treatment would not be as effective as constant treatment.

For macrophage depletion, we implement treatment by removing the fraction of macrophages equal to "Depletion Strength" at the beginning of each treatment cycle (Fig 4A), shown on the *y*-axis. The duration of each cycle is shown on the *x*-axis. Even with cycling treatment on and off, we find that there is a steep increase in the treatment being ineffective to very effective as depletion strength is increased or cycle duration is decreased (Fig 4A-i). The time needed for the treatment to remove the tumor also decreases slightly as depletion strength is increased and cycle duration decreased (Fig 4A-ii). Surprisingly, we see a slight increase in M1 macrophages, total T cells, and active T cells (Fig 4A-iii to 4A-v), especially at the highest depletion strength and slightly longer cycle durations, to a higher level than with constant treatment. This is most likely due to the decrease in IL-4 as tumor cells and M2 macrophages are removed, allowing some of the new macrophages to differentiate into the M1 state.

Whereas macrophage depletion continues to be effective when treatment is cycled, recruitment inhibition becomes rather ineffective, even at complete inhibition of macrophage recruitment (Fig 4B). When cycling recruitment inhibition, treatment was turned on for the number of days shown on the *y*-axis and turned off for the remainder of the cycle (*x*-axis). When treatment is given for almost the entire cycle, there is strong tumor removal, however this quickly falls when treatment is not given for as long or when cycle duration increases (Fig 4B-i). The time needed to remove the tumor is fairly consistent in cases where there is tumor removal (Fig 4B-ii). While there are little to no M1 macrophages for almost all of the treatment combinations, there is one instance where is a large number of M1 macrophages (Fig 4B-iii). However, because that combination saw very little tumor removal, we assume it is a very rare stochastic occurrence. The numbers of T cells and active T cells remain low and constant across combinations, except the instance where there was an increase in M1 macrophages (Fig 4B-iv,v).

Interestingly, a PI3K inhibition of 0.8, which is the lowest successful inhibition for continuous treatment, is still very successful when cycling treatment, even when treatment is given briefly for long cycle durations (Fig 4C-i). For the bulk of the treatment combinations, the time needed to remove the tumor is fairly constant (Fig 4C-ii). M1 macrophages, T cells, and active T cells are all at their maximum values (Fig 4C-iii to 4C-v). To determine why PI3K inhibition continues to be successful even when given for short durations, we examined the time courses when treatment was given for 2 days out of a 25-day cycle (Fig 5). Interestingly, treatment cycles do not always remove the tumor; however, when tumor elimination does happen, it occurs very swiftly (Fig 5A). It is unclear why this occurs, though we do see that on cycles that do not lead to removal, there is little change in the number of M2 macrophages. Because the sustained response that leads to tumor removal is due to T cell produced IFN-γ, we believe that not enough T cells are becoming activated during these cycles to cause the sustained response. We have highlighted a single time course in red to make it easier to visualize. Immediately following a successful cycle of treatment, we see a gradual rise in naive macrophages as the tumor is eliminated (Fig 5B), and a rapid increase in M1 macrophages followed

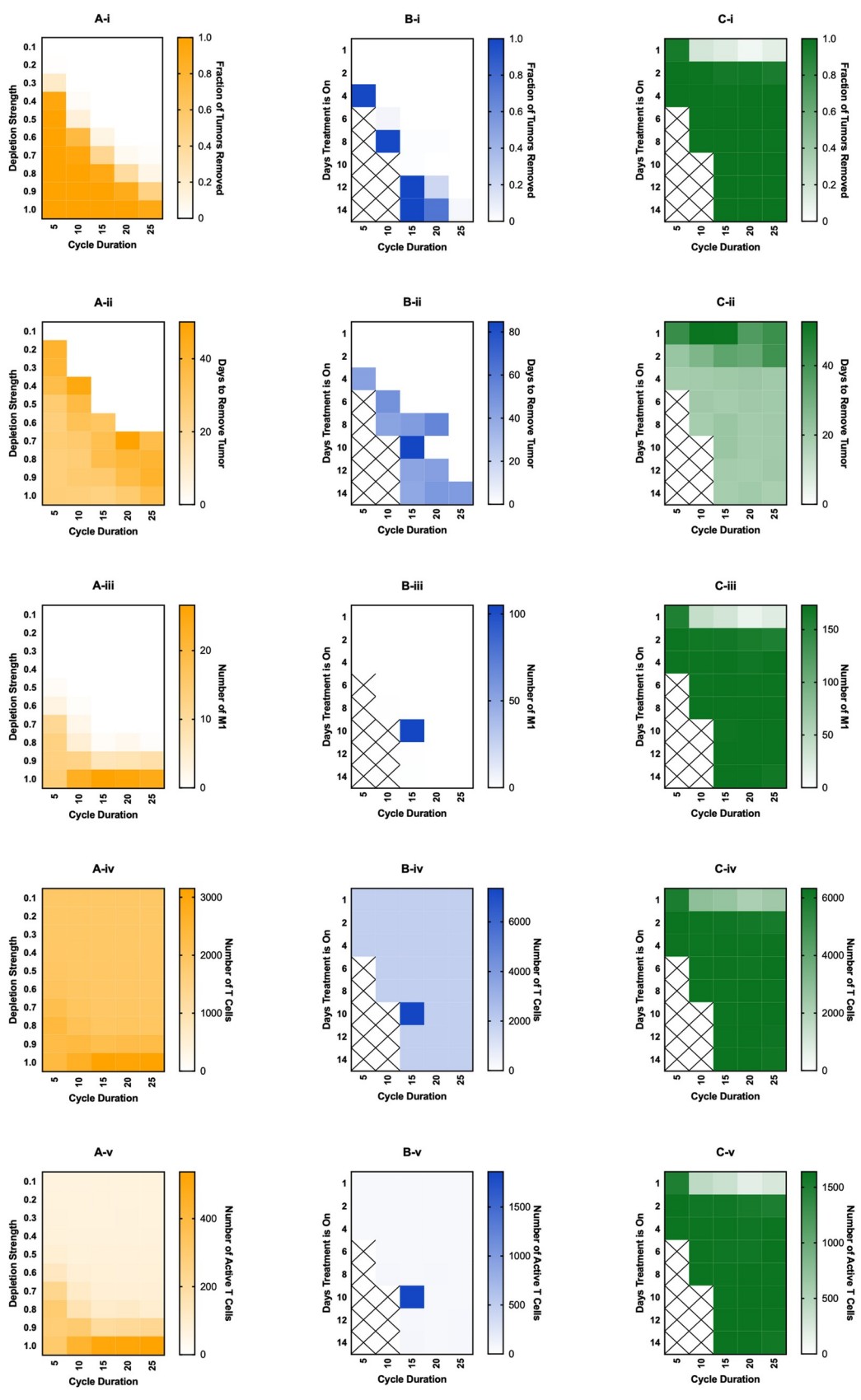

**Fig 4. Effects of cycled immunotherapy started at 100 days of simulation.** For macrophage depletion (A), the fraction of macrophages removed at the beginning of each cycle is given as "Depletion Strength" and the length of each cycle is "Cycle Duration." For recruitment inhibition (B) and PI3K inhibition (C), the number of days in the cycle that treatment is on for is given as "Days Treatment is On." Recruitment inhibition is simulated at a strength of 1.0 (complete inhibition) and PI3K inhibition is simulated at a strength of 0.8. For recruitment inhibition and PI3K inhibition, spaces marked with an X are those where treatment-on time is equal or greater to the cycle duration, thus were not simulated. (i) fraction of tumors removed after starting therapy. (ii) time (days) from starting treatment to tumor removal. It is averaged over the 100 simulations and is equal to zero if no tumors were removed at that treatment level. (iii) maximum number of M1 macrophages. (iv) maximum number of total T cells. (v) maximum number of active T cells.

by a gradual decrease (Fig 5C). M2 macrophages, on the other hand, rapidly decrease since they are being converted in the M1 state (Fig 5D). Corresponding with M1 behavior, the numbers of T cells and active T cells rapidly increase due to influence from the M1 macrophages (Fig 5E and 5F). IL-4 and IFN-γ levels correlate with tumor and T cell dynamics (Fig 5G–5J). We see that due to sustained T cell activation there is sustained IFN-γ, which is responsible for promoting M1 differentiation. This creates a feedback loop between the M1 macrophages and T cells that sustains M1 differentiation and tumor removal even after treatment is removed.

Overall, when cycling treatment on and off, we find that macrophage depletion yields expected results, with higher depletion strengths and shorter treatment cycles leading to a stronger tumor removal. Interestingly, recruitment inhibition becomes very ineffective when cycled, unless treatment is given for almost the entirety of simulation. PI3K inhibition, however, is very effective for almost every combination of time on and time off. We find that this is because converting the macrophages to the M1 phenotype promotes T cell activation and IFN-γ secretion, which sustains the M1 phenotype after treatment is removed.

## Changing tumor proliferation rate and macrophage recruitment rate

Of interest is how tumor proliferation rate and macrophage recruitment rate affect the immune response with and without treatment. Presumably these two parameters would lead to a different equilibrium state, which would change the effectiveness of each treatment. We repeated the above analysis for three sets of tumors: increased proliferation rate, increased macrophage recruitment rate, and both. We first consider tumor growth without treatment, and then implement the three macrophage-based treatment strategies.

Compared to the baseline parameters, we see only a slight increase in the equilibrium number of tumor cells (Fig 6A) when tumor proliferation rate is increased from 0.8 per day to 1.2 per day. What is most interesting is that at random points throughout the equilibrium state, the immune system will suddenly remove the tumor, a phenomenon not seen with the baseline parameters. As with previous simulations, a decrease in tumor cell population is followed by an increase in naive macrophage population (Fig 6B). For simulations that removed the tumor, we see a rapid increase in M1 macrophages with a corresponding decrease in M2 macrophages (Fig 6C and 6D). The dynamics of total T cells and active T cells follows that of M1 macrophages (Fig 6E and 6F). IL-4 and IFN-γ dynamics follow tumor cells and T cells, respectively (Fig 6G–6J).

At the increased tumor proliferation rate, we see a more gradual increase in the effectiveness of constant PI3K inhibition (Fig 7A). However, these results are conflated due to propensity of the immune system to spontaneously remove the tumor during the equilibrium state, as evident by the large error bars in Fig 7B. Across inhibition strengths, there is a large amount of M1 differentiation, T cell numbers, and T cell activation (Fig 7C–7E). Cycling PI3K inhibition is successful throughout the different cycle durations, even with the increased tumor proliferation rate (S9C Fig). Macrophage depletion and recruitment inhibition do not show a clear

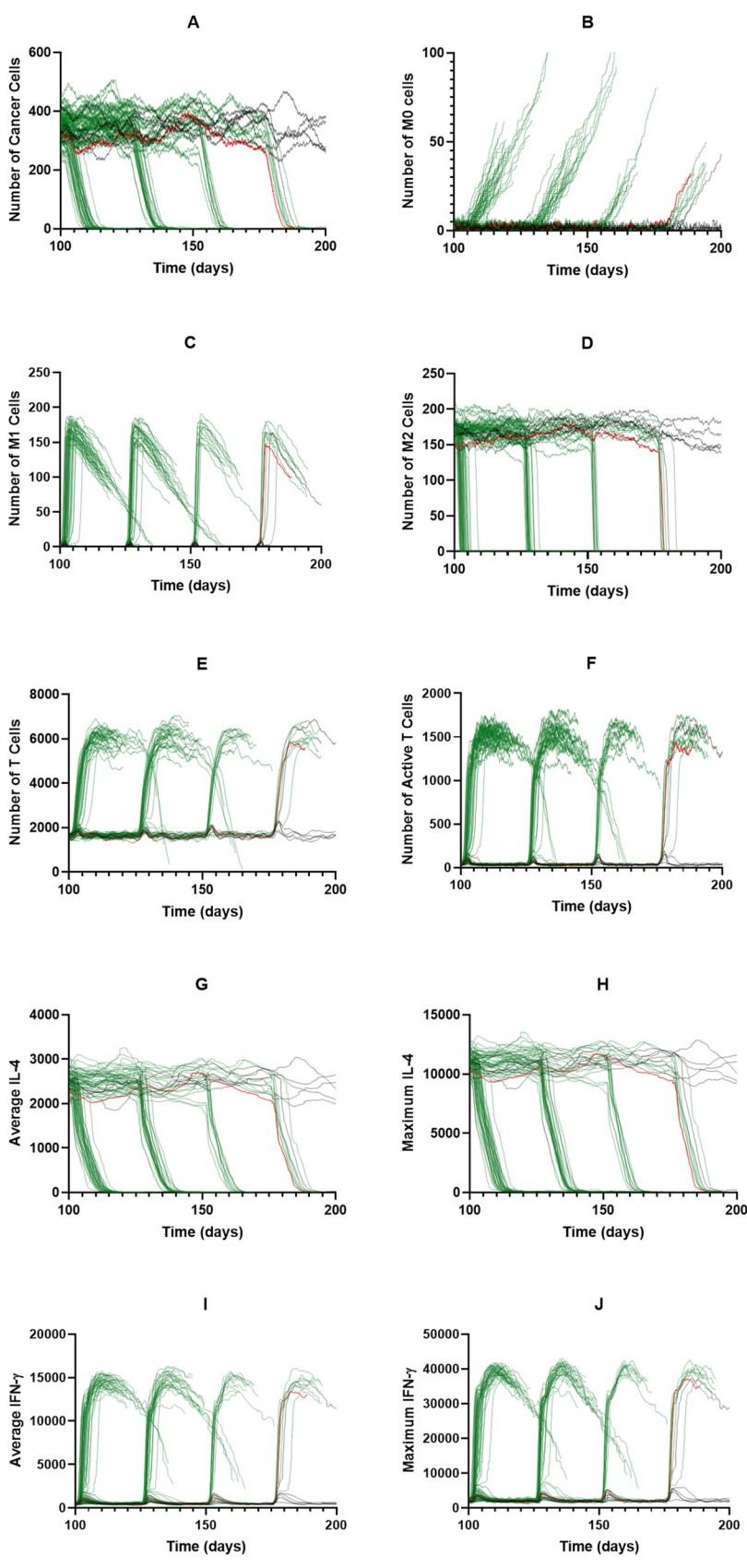

**Fig 5. PI3K inhibition of 0.8 for cycled immunotherapy.** Individual time courses for PI3K inhibition of 0.8 at a cycle duration of 25 days with treatment given for 2 days per cycle. Tumors that survived to the end of simulation are shown in black. Tumors that were eliminated are shown in green. One time course is shown in red for ease of understanding. (A) Cancer cells, (B) M0 macrophages, (C) M1 macrophages, (D) M2 macrophages, (E) T cells, (F) Active T cells, (G) Average IL-4, (H) Maximum IL-4, (I) Average IFN-γ, (J) Maximum IFN-γ.

trend and the results are again conflated due to the inherent tumor removal found without treatment (S9A and S9B Fig, S10 Fig and S11 Fig).

Whereas increased tumor proliferation only caused a slight increase in the equilibrium tumor population, doubling the rate of macrophage recruitment greatly increased the equilibrium tumor population to roughly twice of the baseline (Fig 8A). Naive and M1 macrophage dynamics (Fig 8B and 8C) are very similar to the baseline while dynamics for the remaining cell populations and cytokines reach a higher equilibrium state (Fig 8D–8J).

At the higher macrophage recruitment rate, macrophage depletion therapy bears similar effectiveness as the baseline recruitment rate (Fig 9A), however effectiveness appears at a depletion probability of 0.003 rather than 0.002. The time needed to remove the tumor is constant across depletion probabilities (Fig 9B) and is a bit higher than at the baseline recruitment rate. There is also a large increase in the number of M1 macrophages (Fig 9C), and the peak at a moderate depletion probability becomes more clear. This response is mirrored with the numbers of T cells and active T cells (Fig 9D and 9E), though with a more gradual tail. The other therapies resemble the baseline case (S12, S13 and S14 Figs).

Increasing both tumor proliferation rate and macrophage recruitment rate leads to the highest equilibrium tumor population (Fig 10A). The dynamics of the other cells and cytokines behave similarly to the previous simulations without treatment (Fig 10B–10J).

Interestingly, lower treatment strengths of continuous PI3K inhibition are more effective here than at the baseline parameters, despite the higher equilibrium state of the tumor (Fig 11A), and efficacy increases gradually with inhibition strength. However, at the higher inhibition strengths, treatment is not as effective as it was at the baseline parameters (Fig 3Ci). The time needed to remove the tumor is fairly constant across inhibition strengths (Fig 11B). The numbers of M1 macrophages, total T cells, and active T cells all increase gradually following treatment efficacy, however, due to the wide standard deviations, it does not appear to be too significant of an increase (Fig 11C–11E). Neither continuous macrophage depletion nor recruitment inhibition have a significant ability to remove the tumor (S15 and S16 Figs).

While cycling PI3K inhibition remains similar to the baseline case (S17 Fig), we see a very different behavior with macrophage depletion and recruitment inhibition when tumor proliferation and macrophage recruitment rates are increased, compared to baseline case. In Fig 12 we show the effects of cycling macrophage depletion while the effects of recruitment inhibition, which are very similar, are shown in S18 Fig. Interestingly, moderate depletion strengths and treatment cycle lengths or high depletion strengths at long treatment cycles were the most effective at removing the tumor (Fig 12A). This result is very different from previous simulations, where efficacy correlated to the amount of treatment given. This shows that more moderate treatment could potentially be more effective. While the efficacy here still is not very high, it is higher than continuous treatment for the same tumor parameters. Time needed to remove the tumor is fairly constant for cases where there was tumor removal (Fig 12B). The levels of M1 macrophages and T cells mirror the behavior of tumor elimination (Fig 12C–12E).

Overall, we find that the rate of macrophage recruitment is more impactful on the equilibrium state than the tumor proliferation rate. We also note that, surprisingly, increasing the tumor proliferation rate leads to spontaneous tumor removal during the equilibrium state. As

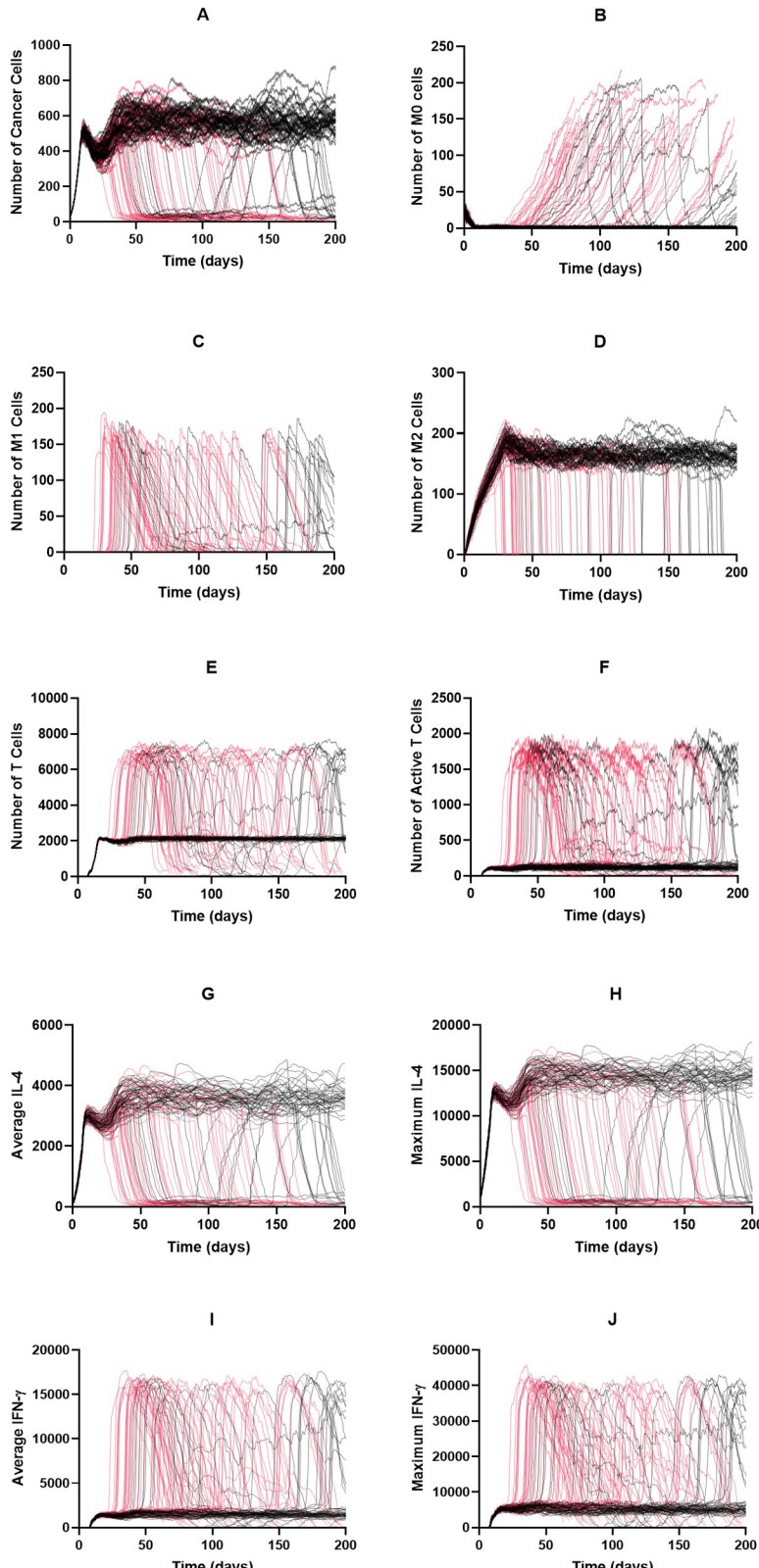

**Fig 6. Time courses with increased tumor proliferation rate and no treatment.** The tumor proliferation rate is increased to 1.2/day. Time courses for tumors that were not removed by the immune system are shown in black; those for tumors that were removed are shown in red. (A) Cancer cells, (B) M0 cells, (C) M1 cells, (D) M2 cells, (E) Total T cells, (F) Active T cells, (G) Average IL-4, (H) Maximum IL-4, (I) Average IFN-γ, (J) Maximum IFN-γ.

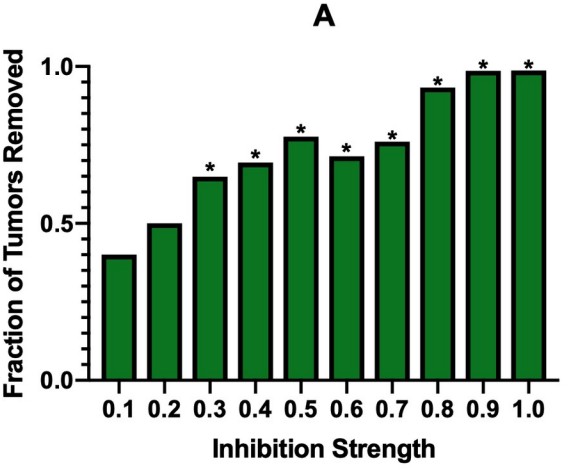

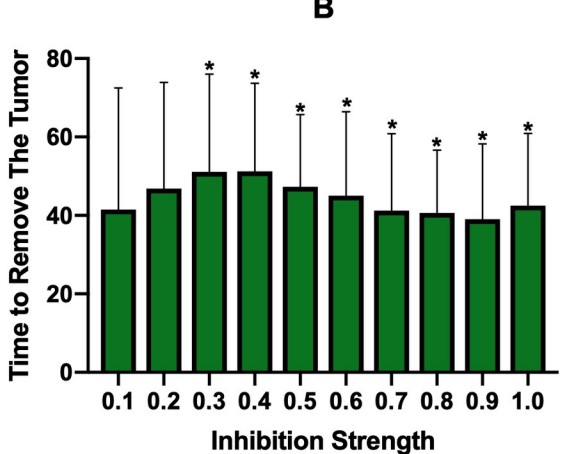

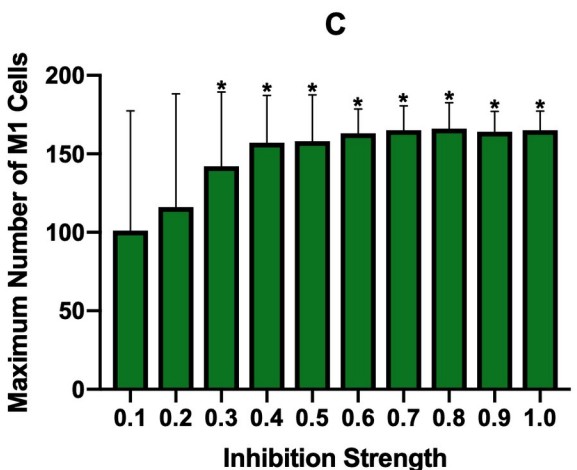

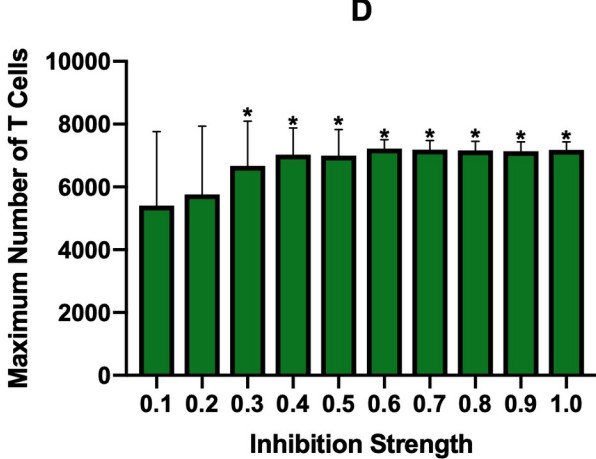

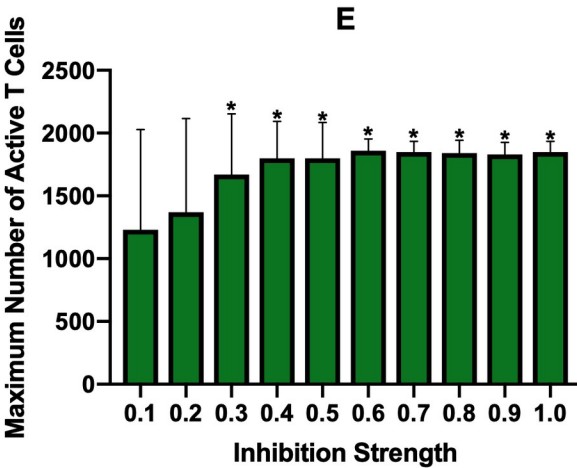

**Fig 7. Constant PI3K inhibition at tumor proliferation rate of 1.2/day.** (A) fraction of tumors removed after starting therapy, (B) average time needed to remove the tumor, (C) the maximum number of M1 macrophages, (D) the maximum number of total T cells, (E) the maximum number

of active T cells. Note the differences in y-axis scales across treatment strategies. Asterisks signify that a result is statistically significant ($p<0.01$) from the result of the lowest treatment strength. We note that for the time needed to remove the tumor (B), plotted is the time averaged over only simulations where the tumor was removed. Therefore, while some bars may appear much higher than that of the lowest treatment strength, they only represent a small number of simulations out of 100 and thus were not found to be statistically significant.

can be expected, treatment efficacy decreases as the equilibrium state increases, though PI3K inhibition retained strong efficacy at higher treatment strengths. The most interesting result is from the simulations at increased tumor proliferation and macrophage recruitment. Here, when cycling macrophage depletion and recruitment inhibition, we find that moderate treatments are actually more effective, causing a slight increase in M1 macrophages and tumor removal, however their effectiveness is well below that of PI3K inhibition.

## Discussion

In this study, we present an ABM examining macrophage-T cell interactions and how macrophage-based immunotherapies can influence tumor growth and the immune response. While macrophages have a number of effects on the TME, including immunosuppression, angiogenesis, and tumor cell invasion [10,11,13,14], we focus on their immune-related interactions and how they can promote or inhibit the T cell response. By using an agent-based model, we are able to explore the emergent behavior that arises from cell-to-cell interactions that would otherwise be very difficult to capture with deterministic equations. To better explore macrophage-based immunotherapies, we utilize a mechanistic model of intracellular macrophage phenotype markers in response to two typical M1 and M2 related cytokines, respectively IFN-γ and IL-4.

We find, consistent with experimental observations, that almost all of the macrophages in the system display an M2 phenotype when no treatment is given, which is indicative of a poor clinical outcome [36]. Interestingly, we find that the system reaches an equilibrium where the T cells are able to function enough to prevent tumor outgrowth but are unable to remove the tumor. This can be considered the "immune control" phase of tumor growth and is very difficult to explore experimentally as *in vitro* methods are unable to capture immune cell recruitment or the long timescale over which immune control occurs. Additionally, it is difficult to find this phase *in vivo* as this phase is completed by the time a tumor can be detected. At our baseline parameters, we find that, at a high enough strength, each treatment has a strong efficacy for removing the tumor when given continuously. When cycling treatment, macrophage depletion retains its efficacy at higher strengths and shorter cycles while recruitment inhibition becomes largely ineffective. PI3K inhibition retains a strong efficacy, even when given for a short amount of time over a long cycle, due to a positive feedback loop between the M1 macrophages and T cells, highlighting the importance of these interactions.

At increased tumor proliferation rates and macrophage recruitment rates, which increase the equilibrium tumor population, macrophage depletion and recruitment inhibition become less effective while PI3K inhibition retains efficacy. What is notable here is that, when cycling macrophage depletion and recruitment inhibition, there is a slight increase in tumor removal and the number of M1 macrophages at moderate treatment cycles, which means that a stronger treatment may not always be the most effective in the case of lowering the number of macrophages in the TME. We believe this is because these treatments remove M2 macrophages from the system, which decreases IL-4 levels and allows T cells to activate, increasing IFN-γ levels. This then allows macrophages newly recruited to the tumor site, which enter the simulation at a distance from the tumor and thus far from peak IL-4 levels, to differentiate to M1 and increase the T cell response. At moderate levels of treatment, enough macrophages are still

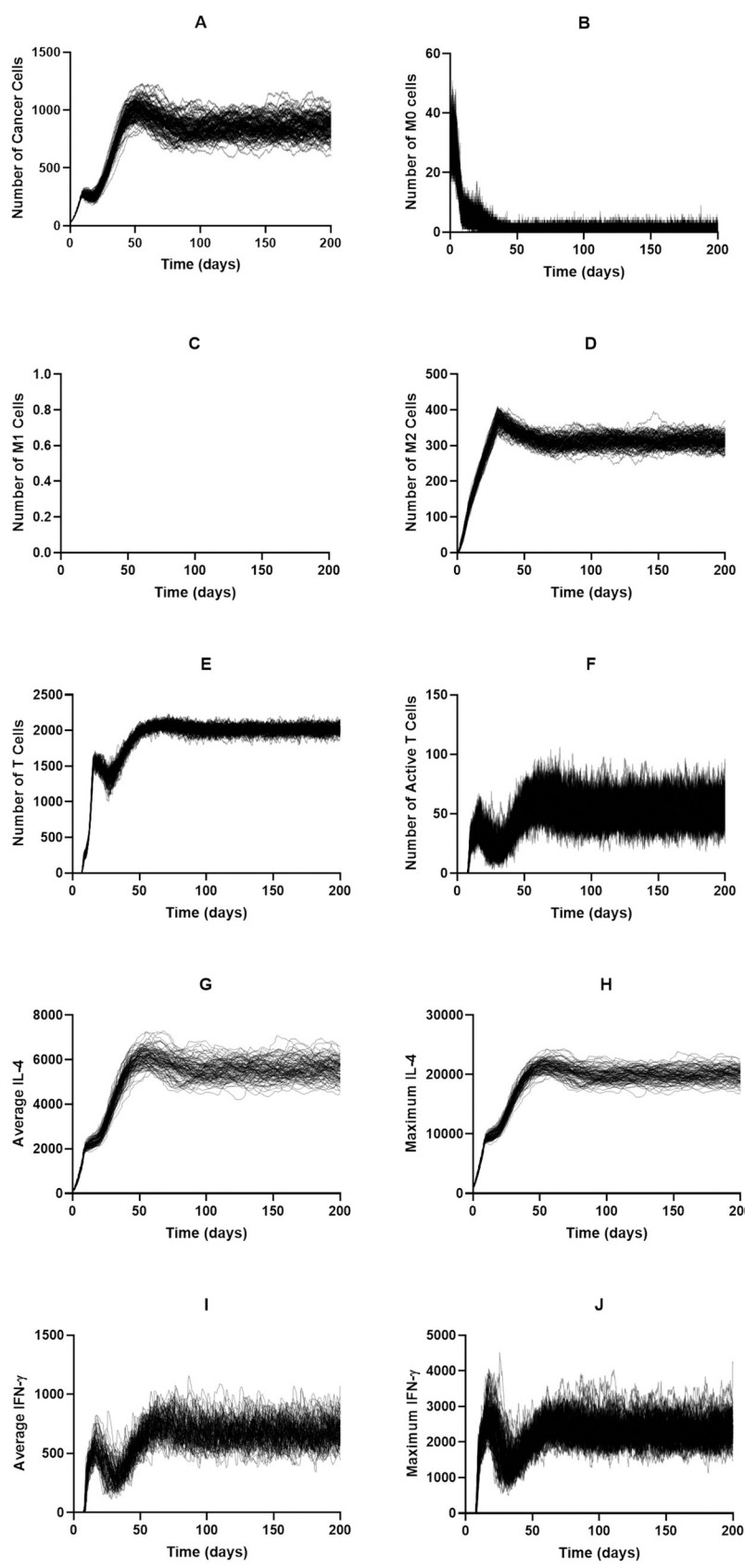

**Fig 8. Time courses with macrophage recruitment rate doubled and no treatment.** (A) Cancer cells, (B) M0 cells, (C) M1 cells, (D) M2 cells, (E) Total T cells, (F) Active T cells, (G) Average IL-4, (H) Maximum IL-4, (I) Average IFN-γ, (J) Maximum IFN-γ.

entering the system to differentiate into M1 macrophages, whereas higher levels of these treatments prevent new macrophages from entering and differentiating. These results, along with the efficacy of PI3K inhibition, highlight the importance of M1 macrophages and their interactions with T cells within the TME. Thus, having a modeling framework that explicitly accounts for these interactions is particularly useful.

While it is difficult to extensively match our simulation results to *in vivo* data, we have found some studies that qualitatively support our results. Nywening et al.[37] present a phase 1b clinical trial of an oral, small-molecule CCR2 inhibitor, which decreases macrophage recruitment to the tumor. They found that this lowers the number of TAMs in the tumor and improves the anti-tumor immune response. This supports our explanation of the model prediction for the effect of reducing macrophage recruitment–that this strategy reduces the number of M2 macrophages (S4 Fig). Germano et al.[38] performed a pre-clinical study examining a macrophage-depleting drug that was able to eliminate macrophages from the tumor site without affecting the number of T cells, which improved T cell removal of the tumor, which is in line with the model predictions (Fig 3A-iv). Kaneda et al.[39] examined PI3K inhibition in mice and found that it stimulates an anti-tumor immune response. They found that, *in vitro*, PI3K inhibition causes a decrease in immunosuppressive macrophage markers and an increase in immune supporting markers. This is similar to the model predictions for how PI3K inhibition affects IL-4 and IFN-γ (S5 Fig). They also found that, in mice, PI3K inhibition led to an increase in IFN-γ expression by T cells and tumor removal.

A novel piece of our work is the reduction of complex mechanistic models into simple neural networks for their inclusion in the individual agents. Though there are several studies that incorporate mechanistic models into ABMs, this brings with it a great computational cost, which is why most models only use simple discrete/stochastic rules. While simplistic rules can still be used to draw great insight about the system, since ABMs are generally concerned about behavior at the multicellular scale, adding this additional biological scale allows us to better understand the mechanistic model and gives us insight into how changes at the intracellular level can compound into changes at the multicellular level. Using the mechanistic model outside of the ABM to train a neural network greatly improves the computational speed, allowing us to run more simulations and explore additional aspects of tumor growth. While some information is inevitably lost, with the neural network only predicting categorical behavior, we believe it to be an acceptable trade-off. We demonstrate that not only can we use the neural network to predict the results of differing cytokine levels, but we can also include different kinetic parameters so that we can simulate the effects of specific targets. This method provides ample opportunity for future simulations where we explore intercellular heterogeneity by including initial protein concentrations in the Monte Carlo simulations and having these protein concentrations as neural network inputs as well.

We acknowledge some limitations of our model. The main limitation is that macrophage phenotype and interactions with T cells are simplified. Macrophages, in reality, display a range of properties and can exist as mixed phenotypes. However, simplifying differentiation into discrete phenotypes, a choice made by similar models, captures enough of the macrophage behavior to be sufficient for this study. Also, the interactions between macrophages and T cells are mediated by many different cytokines, making their interactions much more complex than how we modeled them. However, adding in more cytokines and ligand expressions would

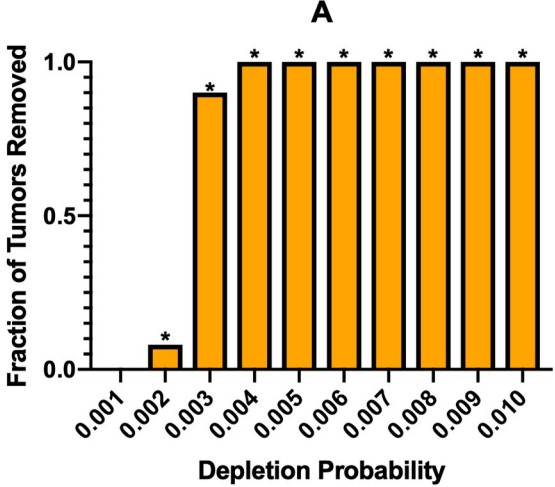

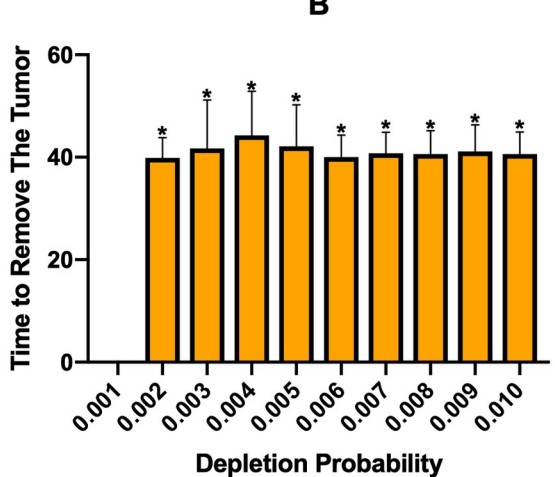

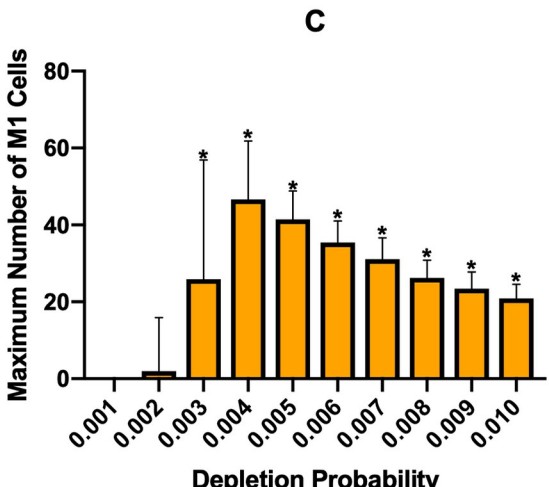

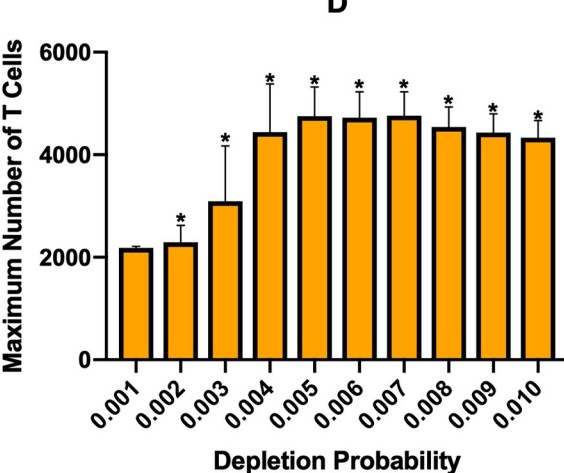

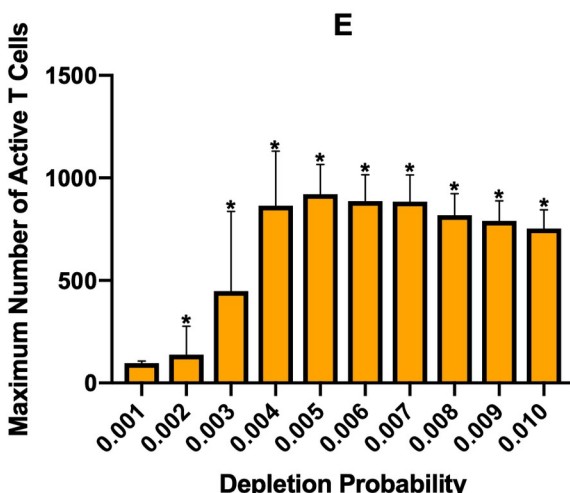

**Fig 9. Constant macrophage depletion started at 100 days with macrophage recruitment rate doubled.** (A) fraction of tumors removed after starting simulation, (B) average time needed to remove the tumor, (C) the maximum number of M1 macrophages, (D) the maximum number of total T cells, (E) the maximum number of active T cells. Note the differences in y-axis scales across treatment strategies. Asterisks signify that a result is statistically significant (p<0.01) from the result of the lowest treatment strength. We note that for the time needed to remove the tumor (B), plotted is the time averaged over only simulations where the tumor was removed. Therefore, while some bars may appear much higher than that of the lowest treatment strength, they only represent a small number of simulations out of 100 and thus were not found to be statistically significant.

have made the model much more complex, and we do not believe this additional layer of complexity would have significantly contributed to the objective of the current study. Another model limitation is that we do not account for nutrient uptake or hypoxia, which promotes M2 differentiation and T cell suppression [31]. We chose to not include these effects because we wanted to focus solely on the interactions between macrophages and T cells. We can account for the effects of nutrients in later models. In addition, in this work, we do not study the effects of changing the T cell numbers or behaviors in order to isolate how macrophage-based strategies influence tumor growth. Expanding the study to vary T cell dynamics can be another focus of future work. We also aim to explore a more detailed spatial analysis of the model to better understand the observed phenomena.

Overall, our model captures the overarching interactions between macrophages and T cells within the TME and predicts how three main macrophage-based immunotherapies impact the immune response to the tumor. We highlight the importance of the interactions between M1 macrophages and T cells for promoting a robust anti-tumor response. We also introduce a method for reducing the computational cost of incorporating mechanistic models into an ABM by training a neural network on the calculated mechanistic model response.

## Methods

Here, we present a multi-scale, hybrid, ABM of the TME, consisting of several cell types and diffusible factors. Our model is lattice-based, with each cell taking up one lattice site. Each cell type has certain behaviors: cells can proliferate into empty lattice sites, migrate to a new lattice site, and interact with other cells and diffusible factors. Using this model, we examine various immunotherapies and their impact on the TME.

### Overview of model

The model represents early tumor growth or a small initial metastasis, and we simulate the tumor in 2D, representing a tissue slice. The model consists of three cell types: T cells, macrophages, and cancer cells, represented as discrete agents. Interactions between cell types and cytokines are shown in Fig 1A. The lattice is a 100x100 grid representing a 1.5x1.5mm tissue slice, with each site being a 15-micrometer square, the size of one cell diameter [40]. As such, only one cell can occupy a site at a time. While different cell types do have different sizes, the "one cell per site" assumption is a necessary limitation of on-lattice models, with many similar models making the same assumption [27,29,30]. Off-lattice models are able to easily account for different cell sizes, however these models are more computationally expensive. Fig 1B displays an example simulation showing the spatial distribution of different cell types. To model the production and diffusion of diffusible factors, we have a layer of partial differential equations (PDEs). Parameters are either taken from previous modeling efforts or set based on experimental observations. Our model does not represent a specific tumor type. Instead, it is meant to examine generalized tumor behavior. We describe the model in detail below and the parameter values are listed in S1 Table, along with supporting references.

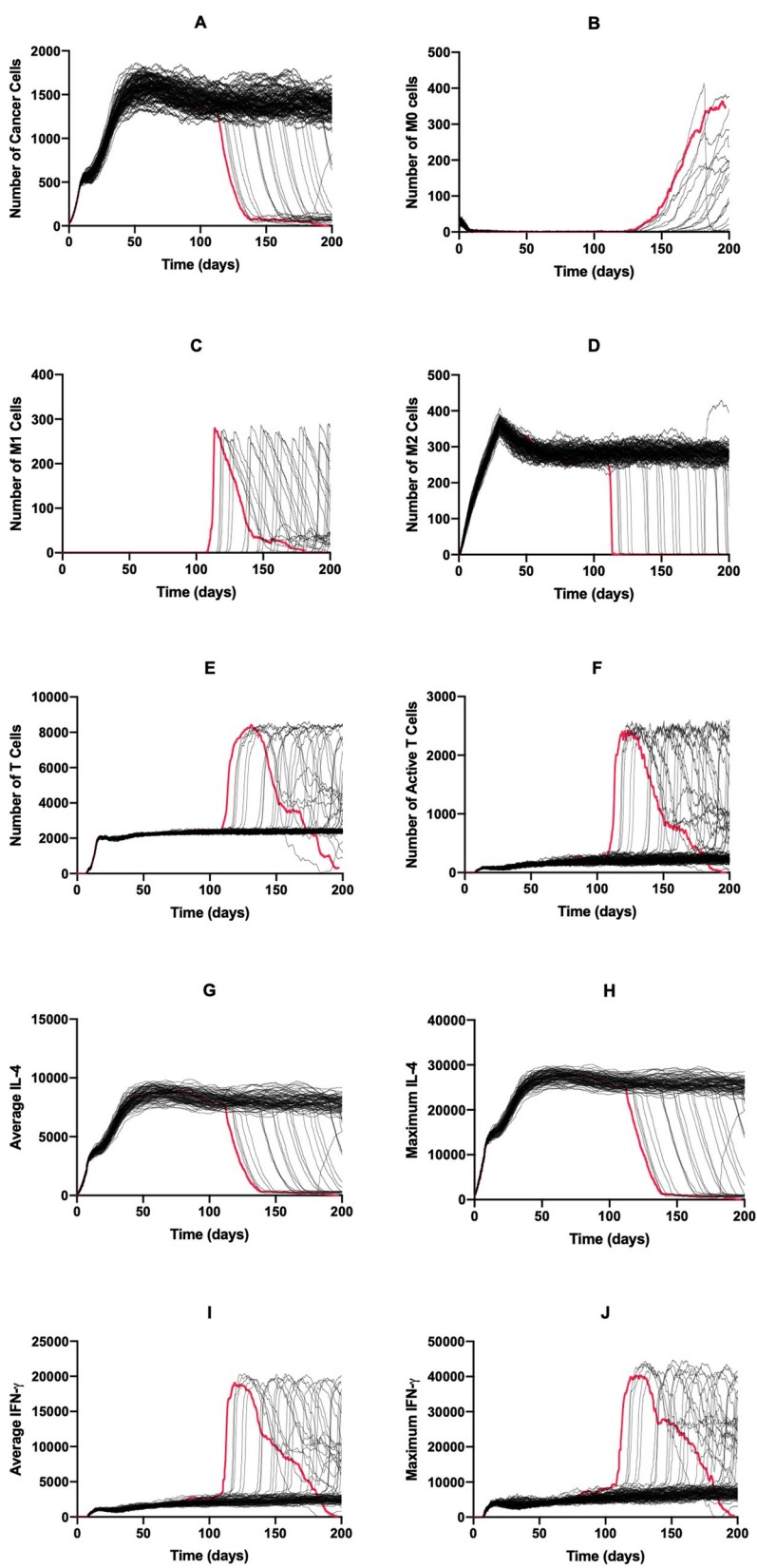

**Fig 10. Time courses with increased tumor proliferation and macrophage recruitment rates, without treatment.**
The macrophage recruitment rate is doubled, and tumor proliferation rate increased to 1.2/day. Time courses for
tumors that were not removed by the immune system are shown in black; those for tumors that were removed are
shown in red. (A) Cancer cells, (B) M0 cells, (C) M1 cells, (D) M2 cells, (E) Total T cells, (F) Active T cells, (G) Average
IL-4, (H) Maximum IL-4, (I) Average IFN-γ, (J) Maximum IFN-γ.

## Diffusible factors

There are three diffusible factors present: a tumor-secreted factor that activates macrophages
and two cytokines, IL-4 and IFN-γ. The first is referred to as the macrophage activation factor,
which alerts the macrophages of the tumor and causes them to differentiate. This factor is
secreted by the tumor cells and primarily represents high mobility group box 1 protein
(HMG-B1) [29]. Also secreted by the tumor, and by M2 macrophages, is IL-4, which is an
immunosuppressive cytokine that promotes macrophage differentiation into the M2 tumor-
promoting phenotype [41]. The final diffusible factor is IFN-γ which is secreted by activated T
cells and is a part of the Th1 response [42]. This immune response promotes macrophage dif-
ferentiation into the M1 immune-promoting phenotype. IL-4 and IFN-γ were chosen because
they are typical pro-tumor and anti-tumor cytokines, respectively. Also, they are used as the
inputs to the mechanistic model used to determine macrophage differentiation, described in
detail in later sections.

The diffusion and secretion of these factors was modeled using PDEs shown in Eq 1.

$$\frac{\partial C_i}{\partial t} = D\nabla^2 C_i + k_{sec,i} Cells(x, y) \tag{1}$$

Here, $C_i$ is the concentration of diffusible factor $i$, $k_{sec,i}$ is the secretion rate of the factor, and
*Cells(x,y)* are the coordinates of cells that secrete the factor. To solve the equations, we use a
finite difference method to discretize them.

## Macrophages

Macrophages are initially present in the tissue, and more are recruited to the TME due to the
secretion of various chemokines, such as CCL2[15,16,43]. To simulate macrophage infiltration
of the tumor, we set a rate of macrophage recruitment [29]. Macrophages enter the TME in
the naïve state (M0). Differentiation occurs once a sufficient level of activating factor is pres-
ent. In our model, a macrophage checks the levels of IL-4 and IFN-γ that are present in their
local environment, and then differentiates according to the intracellular model described in
the following section. Macrophages migrate towards the tumor, mimicking chemotaxis, and
they have a finite lifespan.

## Macrophage differentiation model

To increase the level of biological detail in the model, we incorporate a mechanistic ordinary
differential equation (ODE) model of macrophage intracellular signaling in response to IL-4
and IFN-γ, which was developed by Zhao et al [41]. Although macrophage phenotype is
shown to be on a continuous spectrum [31], we use discrete phenotypes for modeling simplic-
ity, as other papers have done [29,30]. Following the paper by Zhao et al., we first take the
model outputs, which are the time courses for iNOS, TNF-α, CXCL9, CXCL10, and IL-12,
which are characteristic of an M1 phenotype, and IL-10, Arg-1, and VEGF, which are typical
of an M2 phenotype, normalized to their starting values. The product of the time courses of
the M1 outputs is divided by the product of the time courses of the M2 outputs to obtain a
time course of the "M1/M2 Score." While there are many possible ways that macrophage

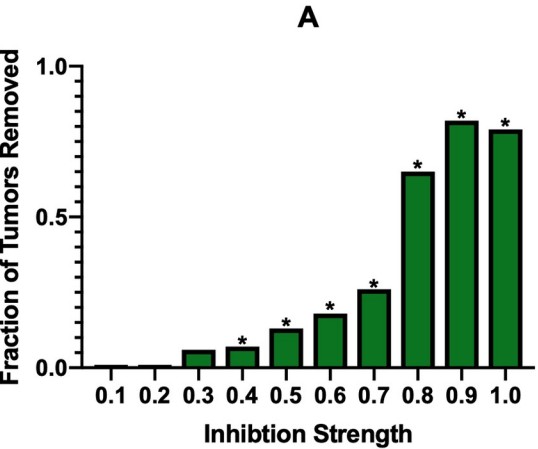

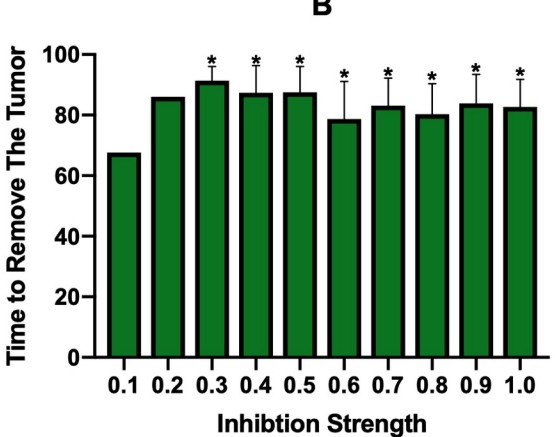

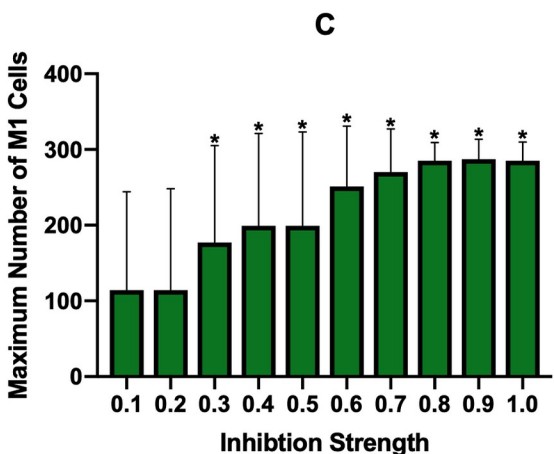

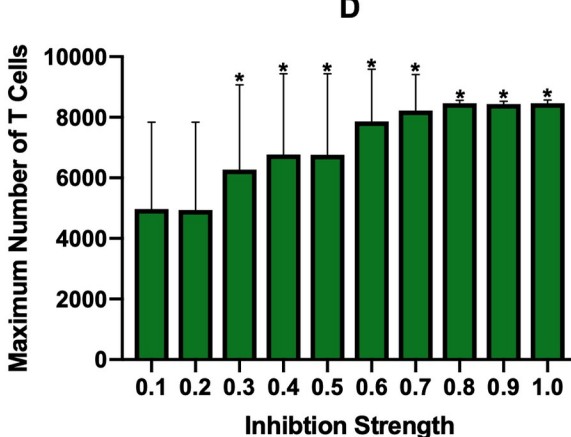

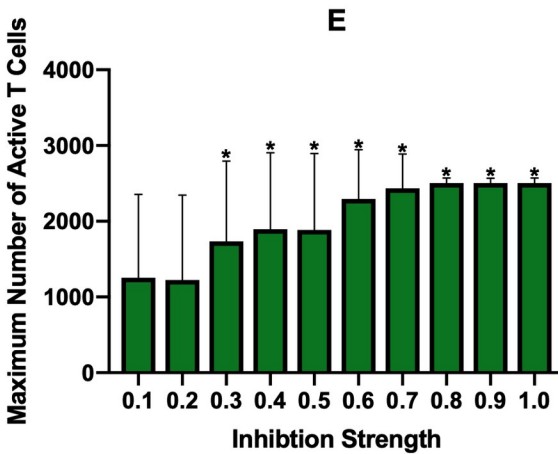

**Fig 11. Constant PI3K inhibition at increased tumor proliferation and macrophage recruitment rates.** (A) fraction of tumors removed after starting therapy, (B) average time needed to remove the tumor, (C) the maximum number of M1 macrophages, (D) the maximum number of total

T cells, (E) the maximum number of active T cells. Note the differences in y-axis scales across treatment strategies. Asterisks signify that a result is statistically significant (p<0.01) from the result of the lowest treatment strength. We note that for the time needed to remove the tumor (B), plotted is the time averaged over only simulations where the tumor was removed. Therefore, while some bars may appear much higher than that of the lowest treatment strength, they only represent a small number of simulations out of 100 and thus were not found to be statistically significant.

differentiation can be determined from here, for our purposes, we calculate this ratio over a 24-hour simulation and then take the average value. If the average value is greater than one, we assume that there is a greater M1-promoting signaling, and the macrophage differentiates into the M1 phenotype. If the value is less than one, the macrophage differentiates into the M2 phenotype. Macrophages are also shown to be plastic, changing their phenotype based on changing environmental conditions [44]. As such, we allow macrophages to reevaluate their local microenvironment, process the input signals via the intracellular signaling model, and redifferentiate every 24 hours. We assume that macrophages can redifferentiate indefinitely.

While ODE models are able to provide an increased level of biological detail to the ABM, they greatly increase the computational burden of simulation. Therefore, to improve computational time, we replaced the mechanistic model with a data-driven model that takes the cytokine concentrations as inputs and outputs the macrophage phenotype. We have already shown that data-driven models are able to predict mechanistic model outputs [45]. Here, we use a neural network to predict macrophage phenotype. Briefly, a neural network is a machine-learning method that predicts outputs given a set of inputs, even with a complex non-linear relationship between the two. It is trained using large sets of input-output data. A neural network can contain several layers, each with many neurons. A single neuron takes as its input the linear combination of the values of either the model inputs or the outputs of the previous layer. The neuron then transforms this input with an "activation function," such as a sigmoid function. The output of this function becomes the neuron's output and is used an input to the next layer.

To train the neural network, we ran 100,000 Monte Carlo simulations of the mechanistic model, randomly sampling cytokine concentrations over the range that they would be present in the ABM. We then determined the phenotype as described above, creating a dataset of inputs and outputs. The neural network was then trained in Python using TensorFlow [46]. The final neural network consisted of one hidden layer with four neurons using a sigmoid activation function. We trained the network several times to determine its ability to capture the ODE model outputs, each time randomly splitting the dataset into training and testing sets. With each testing set, the neural network achieved a prediction accuracy of >98%. Because of this very high accuracy in predicting the output of the ODE model, we determined that the neural network is an efficient way to simplify the ODE model into a simple input-output model.

## T cells

In order to eliminate the tumor, T cells have to be recruited to the TME. Upon cancer cell death, tumor antigen is brought to the lymph nodes, initiating an immune response. To model this process, we implement a T cell recruitment rate for each time step, calculated using the following equations, which were taken from Gong et al [26]. The rate at which T cells are recruited is then multiplied by the duration of each time step to get the number of T cells that are recruited at that iteration, and then that many T cells are randomly placed in available sites on the lattice, since we assume there is sufficient vascularization for immune cell recruitment.

$$r(t) = \frac{k_a N_{c,death}(t - t_{delay})r_1}{\frac{1}{k_i} + N_{c,death}\left(t - t_{delay}\right)} \quad (2)$$

**A**

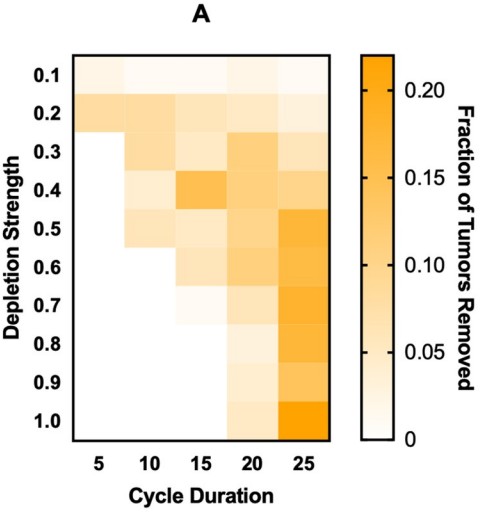

**B**

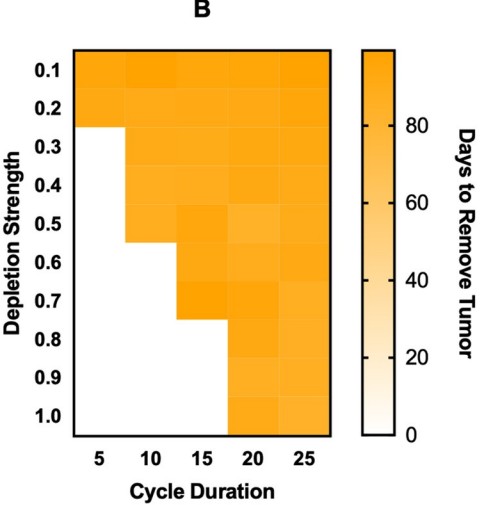

**C**

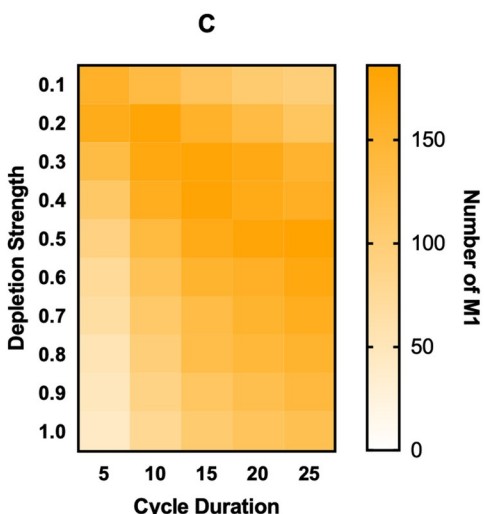

**D**

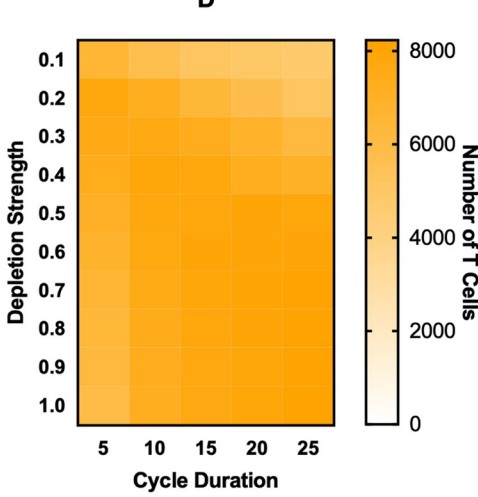

**E**

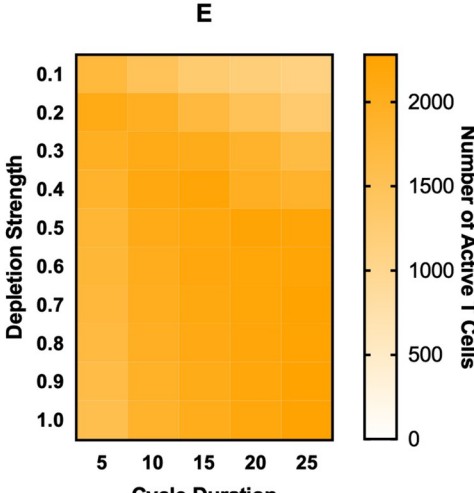

**Fig 12. Cycling macrophage depletion with increased tumor proliferation and macrophage recruitment rates.** The macrophage recruitment rate is doubled, and tumor proliferation rate increased to 1.2/day. (A) fraction of tumors removed after starting simulation, (B) average time needed to remove the tumor, (C) the maximum number of M1 macrophages, (D) the maximum number of total T cells, (E) the maximum number of active T cells.

$$N_{c,death}(t) = \sum_{s=t-0.5*t_{window}}^{t+0.5*t_{window}} n_{c,death}(s) \tag{3}$$

The rate, $r(t)$, of T cell recruitment is calculated using the mutational burden, $k_a$, which is the extent of tumor cell mutation, the basal recruitment rate, $r_1$, the neoantigen strength, $k_i$, and the number of cancer cell deaths over an interval, $N_{c,death}$. While antigen characteristics can vary widely across tumors, we leave them constant as it is not the focus of this study. The $t_{delay}$ parameter accounts for T cell priming and trafficking to the tumor site, leading to a delayed response. The $t_{window}$ parameter sets a time range centered around the current simulation time minus $t_{delay}$ from which to accumulate the number of dead cancer cells, which influences the rate of T cell recruitment.

T cells can do one of four actions: migrate towards the tumor (representing chemotaxis), become fully active upon contact with antigen on a cancer cell, proliferate if fully active, and kill a nearby cancer cell. Fully active T cells are also able to secrete IFN-γ to influence M1 polarization. Interactions between T cells and macrophages are described in more detail below.

## T cell–macrophage interactions

There are numerous ways that macrophages are able to modulate T cell behavior, including cytokine excretion, antigen presentation, and inhibitory ligand expression. To model T cell-macrophage interactions, we condense these behaviors into having macrophages promote or inhibit T cell activation based on their phenotype [9,15,20,47–49]. M1 macrophages are able to activate neighboring T cells, while M2 macrophages are able to prevent neighboring T cells from becoming active. This yields the following equation that determines the probability of a T cell becoming fully active:

$$P_{act} = antigenPresence \times \frac{1}{1 + e^{-k\left(\left(antigenPresence - \frac{numM2}{numM1+1}\right) - s\right)}} \tag{4}$$

where *antigenPresence* equals 1 if there is a tumor cell or an M1 macrophage present to activate the T cell. *numM1* and *numM2* are the number of M1 and M2 macrophages, respectively, neighboring the T cell. The parameters $k$ and $s$ are scaling parameters. We formulated this equation, and hand-tuned parameters $k$ and $s$, to give a reasonable range of probabilities based on neighboring macrophages.

## Cancer cells

Cancer cells in the model simply proliferate. After a cancer cell's internal clock has reached the specified proliferation time, the cancer cell will proliferate, unless there is no room for it to proliferate or it has reached the specified lifespan. If there is not sufficient room for a new cancer cell to be placed, it will become quiescent.

## T cell killing of cancer cells

When an active T cell is neighboring a cancer cell, the T cell can recognize the antigen on the cancer cell and begin to kill it. Due to the time it takes to kill a cancer cell, both the T cell and

the cancer cell are considered to be engaged and do not undergo other processes for the duration of killing [27]. Once the cancer cell dies, it is removed from simulation, and the T cell is free to continue killing, until it reaches the maximum number of cancer cells that it can kill, at which point the T cell becomes exhausted [27].

### Initialization of simulation

Simulation starts with 25 cancer cells placed in the center of the lattice, representing the early growth of a tumor or a micrometastasis. A population of tissue-resident macrophages is randomly scattered throughout the remaining lattice sites. We consider the area surrounding the tumor to be vascularized enough to allow macrophages and T cells to be recruited there.

### Model simulation steps and implementation

At each time step, we proceed through the following steps. First, we calculate the diffusion and secretion of diffusible factors for the duration of the time step. Then, we recruit more macrophages to the environment and proceed to iterate through each macrophage in the environment in a random order and allow them to carry out their various functions. We then repeat this for T cells. After this, we iterate through the cancer cells and allow them to proceed with their functions. Lastly, we remove all dead cells from the environment. The model was implemented in C++. The full model is available at: https://github.com/FinleyLabUSC/Early-TME-ABM-PLOS-Comp-Bio.

### Treatment

As the focus of this study is on interactions between macrophages and T cells, we examine several macrophage-based therapies to see how they impact the ability of the T cells to remove the tumor. For each therapy, we vary the effectiveness, which was implemented as the fraction that the target parameter was reduced by. Treatments were chosen based on targets in the literature, with the two main strategies being to reduce the number of macrophages in the TME and to reeducate M2 macrophages to an M1 phenotype. The three therapies explored here are macrophage depletion, recruitment inhibition, and PI3K inhibition.

Treatment was implemented in two forms. The first was to simulate treatment continuously for the duration of the simulation. The second was to cycle the treatment, with treatment on for several days, then off for several days.

### Recruitment inhibition

As described above, macrophages are recruited to the TME by chemokines such as CCL2. As many macrophages in the TME display an immunosuppressive M2 phenotype, it is thought that preventing macrophage recruitment would allow T cells to better be able to remove the tumor [15,16]. We implement targeting macrophage-recruiting chemokines by reducing the rate at which macrophages are recruited to the environment.

### Macrophage depletion

Similar to recruitment inhibition, depletion of macrophages can be used to reduce the number of macrophages in the TME [15,16]. It is to be noted that both of these strategies fail to discriminate between M1 and M2 macrophages, thus also inhibiting the immune-promoting properties of M1 macrophages. We model macrophage depletion by giving the macrophages a probability of undergoing apoptosis at each timestep that treatment is on. When simulating continuous treatment, we decreased the range of depletion probabilities until we reached the

point where the lowest probability fails to remove a large number of tumors so that we could compare ineffective levels of depletion to effective levels.

## Macrophage reeducation

In order to prevent the elimination of immune-promoting M1 macrophages, reeducation of M2 macrophages into an M1 phenotype is a potential therapy. One noted target is inhibition of PI3K[15,16]. As this is present in the mechanistic model that was utilized for macrophage differentiation, we include variations in PI3K activity in the Monte Carlo simulations and have PI3K activity as an input to the neural network. Targeting this parameter allows us to redifferentiate the macrophages into an M1 phenotype. We simulate this treatment by reducing the value of the PI3K activity parameter.

## Supporting information

**S1 Fig. Results from the Latin Hypercube Sampling–tumor removal.** The fraction of tumors removed as a fraction of relevant tumor microenvironment parameters: macrophage recruitment rate, tumor IL-4 secretion rate, T cell IFN-γ secretion rate, tumor proliferation rate, M2 IL-4 secretion rate, and macrophage lifespan. With Latin Hypercube Sampling, parameter sets are sampled so that each parameter value appears only once. This is done for the sake of computational efficiency. Because only one parameter set was simulated for each parameter value, the plots are very discontinuous and do not show the average model behavior for each parameter value. Despite this, a clear trend is visible for macrophage recruitment rate.
(TIF)

**S2 Fig. Results from the Latin Hypercube Sampling–final tumor cell count.** The final number of tumor cells is on the y-axis and the relevant parameters are on the x-axes (macrophage recruitment rate, tumor IL-4 secretion rate, T Cell IFN-γ secretion rate, tumor proliferation rate, M2 IL-4 secretion rate, and macrophage lifespan). Because only one parameter set was simulated for each parameter value, the plots are very discontinuous. However, a clear trend is visible for macrophage recruitment rate and macrophage lifespan. The maximum number of tumor cells here is 5,000 because we chose to end simulation when either maximum simulation time was reached, the tumor was eliminated, or tumor cell count reached 5,000. This is because at that number of tumor cells, there is much less space available for recruited immune cells, so their recruitment inherently decreases due to the nature of the model.
(TIF)

**S3 Fig. Tumor growth curves without immune cells and without immune function.** Comparison of tumor growth curves when there are no immune cells present in the simulation and when immune cells are present but lack function. Because these curves are very similar, we conclude that spatial inhibition is not the cause of the equilibrium state seen without treatment. Simulations were done in replicates of 10.
(TIF)

**S4 Fig. Impact of initial macrophage density.** Comparison of tumor curves and total macrophage curves for differing starting macrophage densities: $2\times10^{-4}$ (blue), $2\times10^{-3}$ (red, value used for rest of the simulations), and $2\times10^{-2}$ (green) cells per site. Each density was simulated in replicates of 10. Plotted are the average time courses for those replicates.
(TIF)

**S5 Fig. Individual time courses for macrophage depletion probability of 0.006 per timestep.** Tumors that were eliminated are shown in orange. (A) Cancer cells, (B) M0

macrophages, (C) M1 macrophages, (D) M2 macrophages, (E) T cells, (F) Active T cells, (G) Average IL-4, (H) maximum IL-4, (I) average IFN-γ, (J) Maximum IFN-γ.
(TIF)

**S6 Fig. Individual time courses for macrophage depletion probability of 0.002 per time-step.** Tumors that survived to the end of simulation are shown in black. Tumors that were eliminated are shown in orange. (A) Cancer cells, (B) M0 macrophages, (C) M1 macrophages, (D) M2 macrophages, (E) T cells, (F) Active T cells, (G) Average IL-4, (H) maximum IL-4, (I) average IFN-γ, (J) Maximum IFN-γ.
(TIF)

**S7 Fig. Individual time courses for recruitment inhibition of 0.7.** Tumors that survived to the end of simulation are shown in black. Tumors that were eliminated are shown in blue. (A) Cancer cells, (B) M0 macrophages, (C) M1 macrophages, (D) M2 macrophages, (E) T cells, (F) active T cells, (G) Average IL-4, (H) Maximum IL-4, (I) Average IFN-γ, (J) Maximum IFN-γ.
(TIF)

**S8 Fig. Individual time courses for PI3K inhibition of 0.8.** Tumors that survived to the end of simulation are shown in black. Tumors that were eliminated are shown in green. (A) cancer cells, (B) M0 macrophages, (C) M1 macrophages, (D) M2 macrophages, (E) T cells, (F) active T cells, (G) Average IL-4, (H) Maximum IL-4, (I) Average IFN-γ, (J) Maximum IFN-γ.
(TIF)

**S9 Fig. Effects of cycled immunotherapy started at 100 days of simulation at higher tumor proliferation.** For macrophage depletion (A), the fraction of macrophages removed at the beginning of each cycle is given as "Depletion Strength" and the length of each cycle is "Cycle Duration." For recruitment inhibition (B) and PI3K inhibition (C), the number of days in the cycle that treatment is on for is given as "Days Treatment is On." Recruitment inhibition is simulated at a strength of 1.0 (complete inhibition) and PI3K inhibition is simulated at a strength of 0.8. For recruitment inhibition and PI3K inhibition, spaces marked with an X are those where treatment-on time is equal or greater to the cycle duration, thus were not simulated. (i) fraction of tumors removed after starting therapy. (ii) time (days) from starting treatment to tumor removal. It is averaged over the 100 simulations and is equal to zero if no tumors were removed at that treatment level. (iii) maximum number of M1 macrophages. (iv) maximum number of total T cells. (v) maximum number of active T cells.
(TIF)

**S10 Fig. Constant macrophage depletion started at 100 days with increased tumor proliferation rate.** (A) fraction of tumors removed after starting simulation, (B) average time needed to remove the tumor, (C) the maximum number of M1 macrophages, (D) the maximum number of total T cells, (E) the maximum number of active T cells. Note the differences in y-axis scales across treatment strategies. Asterisks signify that a result is statistically significant (p<0.01) from the result of the lowest treatment strength. We note that for the time needed to remove the tumor (B), plotted is the time averaged over only simulations where the tumor was removed. Therefore, while some bars may appear much higher than that of the lowest treatment strength, they only represent a small number of simulations out of 100 and thus were not found to be statistically significant.
(TIF)

**S11 Fig. Constant recruitment inhibition started at 100 days with increased tumor proliferation rate.** (A) fraction of tumors removed after starting simulation, (B) average time needed to remove the tumor, (C) the maximum number of M1 macrophages, (D) the

maximum number of total T cells, (E) the maximum number of active T cells. Note the differences in y-axis scales across treatment strategies. Asterisks signify that a result is statistically significant (p<0.01) from the result of the lowest treatment strength. We note that for the time needed to remove the tumor (B), plotted is the time averaged over only simulations where the tumor was removed. Therefore, while some bars may appear much higher than that of the lowest treatment strength, they only represent a small number of simulations out of 100 and thus were not found to be statistically significant.
(TIF)

**S12 Fig. Constant recruitment inhibition started at 100 days with macrophage recruitment rate doubled.** (A) fraction of tumors removed after starting simulation, (B) average time needed to remove the tumor, (C) the maximum number of M1 macrophages, (D) the maximum number of total T cells, (E) the maximum number of active T cells. Note the differences in y-axis scales across treatment strategies. Asterisks signify that a result is statistically significant (p<0.01) from the result of the lowest treatment strength. We note that for the time needed to remove the tumor (B), plotted is the time averaged over only simulations where the tumor was removed. Therefore, while some bars may appear much higher than that of the lowest treatment strength, they only represent a small number of simulations out of 100 and thus were not found to be statistically significant.
(TIF)

**S13 Fig. Constant PI3K inhibition started at 100 days with macrophage recruitment rate doubled.** (A) fraction of tumors removed after starting simulation, (B) average time needed to remove the tumor, (C) the maximum number of M1 macrophages, (D) the maximum number of total T cells, (E) the maximum number of active T cells. Note the differences in y-axis scales across treatment strategies. Asterisks signify that a result is statistically significant (p<0.01) from the result of the lowest treatment strength. We note that for the time needed to remove the tumor (B), plotted is the time averaged over only simulations where the tumor was removed. Therefore, while some bars may appear much higher than that of the lowest treatment strength, they only represent a small number of simulations out of 100 and thus were not found to be statistically significant.
(TIF)

**S14 Fig. Effects of cycled immunotherapy started at 100 days of simulation at macrophage recruitment rate doubled.** For macrophage depletion (A), the fraction of macrophages removed at the beginning of each cycle is given as "Depletion Strength" and the length of each cycle is "Cycle Duration." For recruitment inhibition (B) and PI3K inhibition (C), the number of days in the cycle that treatment is on for is given as "Days Treatment is On." Recruitment inhibition is simulated at a strength of 1.0 (complete inhibition) and PI3K inhibition is simulated at a strength of 0.8. For recruitment inhibition and PI3K inhibition, spaces marked with an X are those where treatment-on time is equal or greater to the cycle duration, thus were not simulated. (i) fraction of tumors removed after starting therapy. (ii) time (days) from starting treatment to tumor removal. It is averaged over the 100 simulations and is equal to zero if no tumors were removed at that treatment level. (iii) maximum number of M1 macrophages. (iv) maximum number of total T cells. (v) maximum number of active T cells.
(TIF)

**S15 Fig. Constant macrophage depletion started at 100 days with increased tumor proliferation and macrophage recruitment rates.** (A) fraction of tumors removed after starting simulation, (B) average time needed to remove the tumor, (C) the maximum number of M1 macrophages, (D) the maximum number of total T cells, (E) the maximum number of active T

cells. Note the differences in y-axis scales across treatment strategies. Asterisks signify that a result is statistically significant (p<0.01) from the result of the lowest treatment strength. We note that for the time needed to remove the tumor (B), plotted is the time averaged over only simulations where the tumor was removed. Therefore, while some bars may appear much higher than that of the lowest treatment strength, they only represent a small number of simulations out of 100 and thus were not found to be statistically significant.
(TIF)

**S16 Fig. Constant recruitment inhibition started at 100 days with increased tumor proliferation and macrophage recruitment rates.** (A) fraction of tumors removed after starting simulation, (B) average time needed to remove the tumor, (C) the maximum number of M1 macrophages, (D) the maximum number of total T cells, (E) the maximum number of active T cells. Note the differences in y-axis scales across treatment strategies. Asterisks signify that a result is statistically significant (p<0.01) from the result of the lowest treatment strength. We note that for the time needed to remove the tumor (B), plotted is the time averaged over only simulations where the tumor was removed. Therefore, while some bars may appear much higher than that of the lowest treatment strength, they only represent a small number of simulations out of 100 and thus were not found to be statistically significant.
(TIF)

**S17 Fig. Cycling PI3K inhibition with increased tumor proliferation and macrophage recruitment rates.** The macrophage recruitment rate is doubled, and tumor proliferation rate increased to 1.2/day. (A) fraction of tumors removed after starting simulation, (B) average time needed to remove the tumor, (C) the maximum number of M1 macrophages, (D) the maximum number of total T cells, (E) the maximum number of active T cells
(TIF)

**S18 Fig. Cycling recruitment inhibition with increased tumor proliferation and macrophage recruitment rates.** The macrophage recruitment rate is doubled, and tumor proliferation rate increased to 1.2/day. (A) fraction of tumors removed after starting simulation, (B) average time needed to remove the tumor, (C) the maximum number of M1 macrophages, (D) the maximum number of total T cells, (E) the maximum number of active T cells
(TIF)

**S1 Table. Model Parameters.** Parameter values, with supporting references. For the tumor division time and macrophage recruitment rate parameters, values shown in parentheses are the increased values used for certain simulations, as described in the Results.
(DOCX)

## Acknowledgments

The authors thank members of the Finley research group for critical comments and suggestions.

## Author Contributions

**Conceptualization:** Colin G. Cess, Stacey D. Finley.

**Formal analysis:** Colin G. Cess.

**Investigation:** Colin G. Cess.

**Resources:** Stacey D. Finley.

**Supervision:** Stacey D. Finley.

**Writing – original draft:** Colin G. Cess.

**Writing – review & editing:** Colin G. Cess, Stacey D. Finley.

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
