## [Decision Letter · Decision Letter 0]

17 Sep 2020

Dear Dr. Finley,

Thank you very much for submitting your manuscript "Multi-scale modeling of macrophage – T cell interactions within the tumor microenvironment and impacts of macrophage-based immunotherapies" for consideration at PLOS Computational Biology.

As with all papers reviewed by the journal, your manuscript was reviewed by members of the editorial board and by several independent reviewers. In light of the reviews (below this email), we would like to invite the resubmission of a significantly-revised version that takes into account the reviewers' comments.

The reviewers all agree that the work presents an innovative model to address the significant problem of cancer immunotherapy. The reviewers further agree that the results provide useful insights into treatment outcomes, and that overall the paper is well written. However, each individual reviewer raises different comments and questions about specific aspects of the data and figures, and therefore we ask that you address these as appropriate in a revision.

We cannot make any decision about publication until we have seen the revised manuscript and your response to the reviewers' comments. Your revised manuscript is also likely to be sent to reviewers for further evaluation.

Sincerely,

Kathryn Miller-Jensen

Associate Editor

PLOS Computational Biology

Jason Papin

Editor-in-Chief

PLOS Computational Biology

Reviewer's Responses to Questions

**Comments to the Authors:**

Reviewer #1: This well-written paper presents an interesting modeling study of macrophage and T cells interactions in the tumor microenvironment. The methods are clearly explained and the conclusions follow from the results. However, the figures and the figure captions seem to be mismatched.

o Fig 1: caption mentions green dots (M0 macrophages), but figure does not seem to have green dots. The figure, however, seems to show blue dots.

o Fig 2: caption mentions red for tumors removed, but there seems to be no red in the figure (or perhaps it is barely visible?)

o Fig 3: caption mentions A, B, and C, as well as i, ii, iii, and iv, but none of these labels are in the figure. Further, most of the results do not seem to show statistically significant differences. Could the results perhaps be summarized more succinctly?

o Fig 5: caption mentions tumors that survived to be in black color, yet none are shown in the figure. Further, many of the orange tumor sequences seem to be essentially the same as in Fig 4. Is there a need to show Fig 5 or can the results be stated in words?

o Fig 6: caption mentions A, B, and C, as well as i, ii, iii, and iv, but none of these labels are in the figure. Further, the panels do not seem to match what the caption is saying (or if it is, it seems confusing). The units for the colors should be given.

o Fig 9: caption mentions A, B, and C, as well as i, ii, iii, and iv, but none of these labels are in the figure. Most panels seem to show non-significant differences.

o Fig 12: red lines are barely visible.

o Fig 13: units for color scales should be given

Lastly, for those results that are statistically significant, comparisons with appropriate statistics would help to interpret the differences, especially given the stochastic nature of the studies.

Reviewer #2: Reproducibility report has been uploaded as an attachment.

Reviewer #3: In this study, the authors develop a hybrid, multi-scale agent-based model of the tumor microenvironment (TME) that includes cancer, macrophage, and T-cell agents and three soluble factors. The model captures complex interactions and exhibits emergent dynamics without overly complicated descriptions. Overall, this study is elegantly simplistic, in that it asks and answers specific questions, which the model is no more or less complex than it needs to be. This work generates useful and practical conclusions about potential treatments with these therapies, and it builds nicely upon prior work in this area. This work is likely to be of interest to the systems biology community, as well much broader cancer immunotherapy audiences. There are several overall and specific concerns that should be addressed prior to publication:

1. The manuscript would benefit from additional discussion, speculation, or insight to help interpret some of the unintuitive results, such as why moderate treatment was preferred in cases where tumor proliferation and macrophage recruitment rates were both increased.

2. Please be more explicit when supporting the claim that previously reported experimental results are comparable to the results found here. How do those experimental setups compare to the model used here? Did these experiments take place in a tissue with vasculature or in a dish? Were there nutrient limitations? Please describe exactly which trends and observations are used as model validation or verification in this study.

3. The use of the neural net to avoid explicitly model intracellular signaling is innovative and interesting. However, no data nor analysis is presented to validate this model simplification; this gap should be addressed.

4. One limitation of the model seems to be the lack of accounting for nutrients or nutrient limitations, which cause common features of tumors such as hypoxia and necrotic cores. While the authors do not necessarily need to add this phenomenon to their model, this should be discussed as a limitation, and artifacts arising from this choice should be considered.

5. Related to the last point, it would be worth investigating whether the factors that limit tumor growth are truly immunological control vs. nutrient limitation, space limitation, etc. It does not seem obvious that a rather arbitrarily parameterized model would yield a scenario in which immune control is achieved.

6. If each lattice site fits one agent, this is concerning because cancer cells have a far larger volume than do T-cells, and it is possible that this could impact both replication and motility. The authors must provide argument or evidence that this isn’t a problem.

7. It would be interesting to see an analysis of the spatial distribution of the cell types throughout the simulation. Treating solid tumors is complicated because it can be difficult for immune cells to penetrate the tumor core. An analysis like this would be useful in showing that the model has the capability to collect data that is difficult to collect in vitro or in vivo but that is relevant, important, and interesting.

8. Please clarify how treatment cycling, as modeled here, relates to clinical scenarios of relevance to patient treatment practices. Is this simply a matter of patient compliance or is there more to this phenomenon?

9. The authors mention that for constant vs cycled treatment, they change the probability of macrophage recruitment in order to avoid complete depletion of macrophages in the constant treatment case. This raises concerns about making a fair comparison between continuous vs cycled treatment, as this probability change introduces bias. Please justify why this comparison is fair; the given statement “If it were as the same strength in both cases, the constant treatment would basically get treated immediately” is insufficiently precise.

10. The authors report “data not shown” for a few data sets, concerning treatment of specifically altered types of tumors; these data are interesting and should be included in the supplement.

11. In one part of this study, there seems to be a gap in the analysis presented. It would be helpful to evaluate PI3K treatment efficacy in the case where only macrophage recruitment rate in the tumor was changed, since this treatment was evaluated for tumors with only tumor proliferation rate altered or with both tumor proliferation rate and macrophage recruitment rate changed. This, the suggested evaluation seems like a missing case/control.

12. Showing the PI3K inhibition data for tumors with both tumor proliferate rate and macrophage recruitment rate altered (Fig 9B) far before discussing these data is confusing; it might be more clear if these data were moved to later to match the text.

13. The figures and data presentation could be substantially improved by considering a few general suggestions:

a. Include labels within figures to annotate panels/columns and scale bars (i.e., tumor removal, cancer cell count, etc.) to indicate what is being measured without requiring the reader to consult the figure caption

b. Normalizing scale bars and graph axes that measure the same thing across treatment types would make it easier to compare outcomes across treatments

14. In general, presenting simulation results as a series of trajectories provides a nice way to view the distribution of outcomes within any one panel. However, it does not enable one to evaluate the correlations/relationships between the data presented in any two (or more) panels. Would it be possible to include more rigorous calculations to evaluate such relationships (e.g., to back up or test statements such as “outcome X tends to occur when behavior Y is observed”, or to evaluate positive and negative correlations described in the text)?

15. Have you considered running a simulation without immune cells to evaluate the baseline dynamics of tumor size and immune-independent phenomena? This base case could enable comparison with some experimental systems that match this scenario.

Specific comments:

Line 56: Please clarify (up front) that macrophage functional polarization, while useful at a conceptual level, actually represents a continuum of phenotypes rather than discrete M1 vs M2 categorical distinctions. This is discussed at the end the paper but should be noted here, too.

Line 72-73: The authors note one cause of T-cell exhaustion. However, for completeness it may be useful to mention that T-cell exhaustion is also caused by excessive and continuous stimulation, not just checkpoint proteins.

Wherry, E. J., & Kurachi, M. (2015). Molecular and cellular insights into T cell exhaustion. Nature Reviews Immunology, 15(8), 486–499. https://doi.org/10.1038/nri3862

It is possible that the phenomenon of exhaustion is captured by M2 macrophages preventing neighboring T-cells from becoming active. What happens to T-cells after they reach their 5 kill max limit? Is that supposed to be T-cell exhaustion? The authors should be clearer about how/whether their model describes T-cell exhaustion.

Line 80: It could be worth noting that more recent findings suggest that blocking inhibitory signals does not actually recover exhausted T-cells but it might help new T-cells infiltrate the tumor and prevent them from becoming exhausted.

Yost, K. E., Satpathy, A. T., Wells, D. K., Qi, Y., Wang, C., Kageyama, R., … Chang, H. Y. (2019). Clonal replacement of tumor-specific T cells following PD-1 blockade. Nature Medicine, 25(8), 1251–1259. https://doi.org/10.1038/s41591-019-0522-3

Line 116: Consider noting and discussing the following ABM paper, which includes many cell types interacting at a site of infection (not necessarily a tumor, but might be worth noting)

Folcik, V. a, An, G. C., & Orosz, C. G. (2007). The Basic Immune Simulator: an agent-based model to study the interactions between innate and adaptive immunity. Theoretical Biology & Medical Modelling, 4, 39. https://doi.org/10.1186/1742-4682-4-39

Line 165: Figure S1: Please elaborate upon the statement that the parameters besides macrophage recruitment rate seem to have sharp peaks in seemingly random places; why might this be the case?

Line 170: “TAMS” should be “TAMs”

Line 182: Please briefly explain why T cells are introduced at a specific timepoint (this is also not clear from the model description).

Line 182: 350 cells seems like a very small tumor. What is the total area/volume this tumor covers? 4000 molecules of IL-4 also seems somewhat small? Given the number of macrophages, it seems like this would be higher? The authors should consider justifying the number of each cell types or at least relative fractions of each that the model finds as an outcome as part of model validation.

Line 202: “effects” should be “affects” since it’s the verb

Line 213: To clarify, does this treatment remove macrophages of each kind (M0, M1,M2) with equal probability? This was later discussed but it would be useful to clarify here where it is defined.

Line 214: “s-axis” should probably be “x-axis”

Line 213-225: These data require a bit more investigation. It is not clear how increasing the depletion probability from 0.001 to 0.002 yields a dramatic increase in tumors removed while conferring no discernible impact on the max number of M1 cells or the max number of activated T cells. The authors appropriately avoid drawing a strong conclusion, but even the general mechanism proposed does not explain this initial sharp rise. Is it possible that this phenomenon arises due to competition for space on the lattice?

Line 219: Can this be called biphasic, given the error? The term biphasic is a bit too strong of a claim and doesn’t add to your argument.

Line 282: Could you do a spatial analysis to try to prove or disprove this claim? For example, is it possible (or useful) to compare spatial distributions of each cell type between cases where the tumor is eliminated vs. not?

Line 284: Why do you think that no M1s appear here in this treatment but do appear in the depletion treatment if both treatments have the overall effect of lowering the number of macrophages in the simulation? Please clarify this reasoning.

Line 291: The claim that PI3K inhibition showed the fastest removal time seems to be too strong as that only seems to be the case for one of the bars shown; otherwise, column B seems to have faster removal times on average for cases where the treatment was effective. However, this comparison is difficult to make visually because the axes all have different ranges. Please scale the axes to have the same range and max (across rows) for easier visual comparison (or at least in all the ii graphs).

Line 418: Where is the data for cycling PI3K inhibition? Please add a figure citation.

Line 422: Where is the data for PI3K inhibition for tumors with macrophage recruitment rate doubled and unchanged tumor proliferation rate? Please add a figure citation.

Line 445: The claim that this response in 11C is biphasic does not seem justified, as it seems too strong. It could easily be a left-shifted single peak.

Line 456: In the sentence “behave similarly to that case”, to what does “that” refer? This behavior appears to be a mix of the behaviors seen in the increased tumor proliferation only case and the increased macrophage recruitment rate only case.

Line 473: These data would be useful to show in the supplement, since a key goal of this paper is investigating how these treatments work across tumor types (i.e., tumors with varying parameters).

Like 497: It would be nice to show this result (treatment efficacy vs. equilibrium state) on a graph for each treatment type.

Line 617: How many macrophages are initialized in the tumor at the start of a simulation? This is not clearly defined in the methods and should be for the purpose of reproducibility and clarity. Additionally, if the authors have not done so already, please consider evaluating how changing this number impacts treatment outcomes.

Fig. 1B: It is very hard to see the green color with the gray background. Please also consider recoloring to accommodate readers who are red/green color blind.

Fig. 3: It seems that “Inhibition Strength” on the x-axes in Fig 3C’s are different than the “Inhibition Strength” on Fig 3B’s, but the authors might want to indicate that in some way (ex: “PI3K Inhibition Strength”); it’s currently confusing. This could be further clarified by the general suggestion to add labeling within figure/columns.

Fig. 6: Please label the scale bars on the figure. It would be useful to scale the color bars which depict the same quantity to be consistent across any one row.

Fig. 7: Why are there distinct groups of tumors that respond in qualitatively similar ways but at different times? The authors should attempt an explanation of this observation, and it would be interesting even if it were simply speculation.

Reviewer #4: This paper describes the development and use of an agent-based computational model to explore novel macrophage-based immunotherapy strategies for treating cancer. The paper is very well written and the data support the conclusions. The topic of cancer immunotherapy is timely and the authors’ use of a computational model to offer new insight into therapeutic strategies that do not yet exist is an elegantly practical deployment of this modeling approach that provides the field with a way to move beyond “thought experiment” and into the realm of “quantitative prediction”. The overarching modeling approach are innovative, particularly in its deployment of a neural network to reduce computational complexity of the intracellular signaling component of the simulation. I think readers of this journal will find the paper to be a substantial contribution; however, I have some suggestions for consideration that could elevate the impact and improve the credibility of the model.

1. The title is a bit long and ambiguous...”and impacts of macrophage-based immunotherapies [on what?]”

2. Line 214 refers to “s-axis”, presumably this is a typo and the authors mean “x-axis”.

3. Line 220 the authors state that the biphasic response in the maximum number of M1 macrophages is due to a decrease in IL-4 as tumor cells and M2 macrophages are removed, and it would be helpful to show plots of IL-4 to substantiate this claim.

4. Line 282 refers to spatial stochasticities but it is not clear to what extent the spatial nature of the model impacts the outcomes/predictions.

5. The focus of the paper is on predicting and evaluating the impact of adjusting macrophage dynamics, but a helpful “control” (counter-point) simulation would be to adjust T-cell numbers/dynamics.

6. Line 393 refers to “equilibrium state”, but it is not clear how the authors define this state. Line 496 refers to “equilibrium phase” – is this the same thing as “equilibrium state”?

7. The authors offer no attempt at model validation with independent experimental data and if it were possible to benchmark at least some of the predictions to experiments that would strengthen the credibility of the model.

**Have all data underlying the figures and results presented in the manuscript been provided?**

Reviewer #1: Yes

Reviewer #2: None

Reviewer #3: **No: **Some of the data are listed as “data not shown,” but they should be included, as noted in comments to authors.

Reviewer #4: Yes

PLOS authors have the option to publish the peer review history of their article (what does this mean?). If published, this will include your full peer review and any attached files.

Reviewer #1: No

Reviewer #2: No

Reviewer #3: No

Reviewer #4: No
---

## [Decision Letter · Decision Letter 1]

11 Nov 2020

Dear Dr. Finley,

We are pleased to inform you that your manuscript 'Multi-scale modeling of macrophage – T cell interactions within the tumor microenvironment' has been provisionally accepted for publication in PLOS Computational Biology.

Reviewer 3 had some additional suggestions that I urge you to consider prior to submitting the final version of your paper (comments appended below). However, I do not think these require additional review and so I will leave these revisions to your discretion.

Best regards,

Kathryn Miller-Jensen, Ph.D.

Associate Editor

PLOS Computational Biology

Jason Papin, Ph.D.

Editor-in-Chief

PLOS Computational Biology

Reviewer's Responses to Questions

**Comments to the Authors:**

Reviewer #1: Thank you for addressing the reviewers' feedback.

Reviewer #3: General Comments:

The authors did a thorough job of addressing the concerns in the first round of revisions, and the manuscript has been substantially improved. There are a few minor overall and specific concerns that should be addressed prior to publication:

1. The terms “macrophage reeducation” and “PI3K inhibition” are used interchangeably/inconsistently, which could be confusing to readers. Please use only one of the terms for consistency, or alternatively, be clear as to how these terms are to be uniquely employed. Incidents include:

a. Page 30, line 308

b. Page 32, Line 354

c. Page 34, Line 410

2. The authors mention that when comparing constant vs. cycled treatment, in the constant treatment case *only*, they change the probability of macrophage recruitment in order to avoid complete depletion of macrophages. This therefore seems like this is an unfair comparison between these treatment strategies, because in reality, two variables are being changed—both frequency of treatment and strength of treatment. The response provided by the authors after the first round of review is helpful for understanding the choices made from a computational standpoint, but this does not address the aforementioned concern about the way this comparison is presented.

Specific comments:

Page 19, Line 75 – remove the word “from” after cytokine (if this is a typo, as it seems).

Page 26, Line 221 – this is the first time that “reeducation” has been written with a hyphen. Please be consistent with hyphenation of this term.

Page 33, Line 373 – The authors say that the observation on the heat map (Fig. 4, Biii) is a “rare stochastic occurrence”. This raises the question of how many replicates are represented by the heat map? If it is only one, the authors should do a few and average them. If it is the average of many replicates already, then it doesn’t seem like just a random stochastic phenomenon, but more like an emergent trend. Please clarify which is the case.

Page 46, Line 683 – Insert a comma after the word “IFN-γ” and before the word “which”

Reviewer #4: The authors have adequately addressed my initial concerns.

**Have all data underlying the figures and results presented in the manuscript been provided?**

Reviewer #1: Yes

Reviewer #3: Yes

Reviewer #4: Yes

PLOS authors have the option to publish the peer review history of their article (what does this mean?). If published, this will include your full peer review and any attached files.

Reviewer #1: No

Reviewer #3: No

Reviewer #4: No

---

## [Editor Report · Acceptance letter]

9 Dec 2020

PCOMPBIOL-D-20-01387R1 

Multi-scale modeling of macrophage – T cell interactions within the tumor microenvironment

Dear Dr Finley,

I am pleased to inform you that your manuscript has been formally accepted for publication in PLOS Computational Biology. Your manuscript is now with our production department and you will be notified of the publication date in due course.

With kind regards,

Nicola Davies
